# Differential reinforcement encoding along the hippocampal long axis helps resolve the explore–exploit dilemma

Alexandre Y. Dombrovski [1,4], Beatriz Luna[1] & Michael N. Hallquist [2,3,4 ✉]

When making decisions, should one exploit known good options or explore potentially better alternatives? Exploration of spatially unstructured options depends on the neocortex, striatum, and amygdala. In natural environments, however, better options often cluster together, forming structured value distributions. The hippocampus binds reward information into allocentric cognitive maps to support navigation and foraging in such spaces. Here we report that human posterior hippocampus (PH) invigorates exploration while anterior hippocampus (AH) supports the transition to exploitation on a reinforcement learning task with a spatially structured reward function. These dynamics depend on differential reinforcement representations in the PH and AH. Whereas local reward prediction error signals are early and phasic in the PH tail, global value maximum signals are delayed and sustained in the AH body. AH compresses reinforcement information across episodes, updating the location and prominence of the value maximum and displaying goal cell-like ramping activity when navigating toward it.

[1] Department of Psychiatry, University of Pittsburgh, Pittsburgh, PA 15213, USA. [2] Department of Psychology, Penn State University, University Park, PA 16801, USA. [3] Department of Psychology and Neuroscience, University of North Carolina, Chapel Hill, NC 27599-3270, USA. [4]These authors contributed equally: Alexandre Y. Dombrovski, Michael N. Hallquist. ✉email: michael.hallquist@gmail.com

Decisions under uncertainty involve a difficult tradeoff between exploiting familiar valuable options and exploring unfamiliar and potentially superior ones[1]. Much has been learned about the neural mechanisms of exploration in the prefrontal and cingulate cortex[2–5], as well as the striatum and amygdala[6], using reinforcement learning (RL) paradigms with unstructured discrete options. Unlike these paradigms, however, most real-world environments have a complex spatial, temporal, or abstract structure[7,8]. Efficient exploration and exploitation in these settings requires allocentric cognitive maps of the type found in the hippocampus[9], and exploration can be defined not only as sampling of lower-valued options, but also as distance traveled through space. Here, bridging the RL and cognitive mapping literatures, we propose an account of how the human brain resolves the explore/exploit dilemma: posterior hippocampus (PH) invigorates exploratory shifts while anterior hippocampus (AH) supports convergence on the best option.

The hippocampus displays a functional long-axis gradient (see below), dorsal–ventral in rodents and posterior–anterior in primates (hereafter: PH and AH). This domain-general gradient was initially thought of as cognitive-motivational[10] and more recently, as fine-coarse[11–13], an account inspired by the finding that the size of place field representations increases along the long axis (e.g., ref. [14]). Furthermore, while dorsal (posterior) hippocampus rapidly develops representations of specific objects and locations, ventral (anterior) hippocampus gradually learns to identify relationships among objects, locations, and contexts that predict rewards[15]. This rich literature, however, is mostly atheoretical and does not formally distinguish between hippocampal substrates of exploitative, reward-guided actions, and those of exploratory actions that forego short-term rewards.

Researchers have sought to explain how the hippocampus maps rewards using models that rely on reward prediction error (RPE) signals to credit reinforcement to previous states and actions[16,17]. Empirical studies have found that PH is required for this process[18–20], supporting "model-based" learning. RPEs are reported by the dopaminergic mesostriatal pathway. Furthermore, dopaminergic inputs into the dorsal hippocampus from the midbrain[21] and the locus coeruleus (LC)[22] enhance spatial memory for rewarded and salient locations and promote exploratory behavior[23]. RPEs have also been found in rat dorsal CA1[24]. Likewise, a handful of human imaging studies[25,26] find RPEs in the PH, though PH is not prominent in imaging meta-analyses (see Supplementary Fig. 1 and refs. [27,28]).

AH, by contrast, responds to global features of reinforcement: whether the environment is aversive[29], whether one is approaching the goal[30–32], and whether one is at the location of the preferred reward in the environment[33]. Whereas PH is critical for the development of cognitive maps that support allocentric navigation, AH supports behavioral flexibility in reaching the goal[34]. In addition, the AH is preferentially connected with ventromedial prefrontal cortex (vmPFC) (rodent prelimbic cortex), which represents abstract reward value, and this connectivity is important for motivated behavior[29]. One resource-efficient way to map the goal is to compress value representations by selectively maintaining values of preferred actions and forgetting inferior alternatives. We have found that this compression strategy facilitates the transition from exploration to exploitation[35]. AH, having access to values of remote states and carrying coarse representations, may implement such compression to track global statistics of the environment.

Altogether, we hypothesized that PH and AH play functionally dissociable roles in exploration and exploitation, respectively. PH holds detailed, concrete representations of specific states and invigorates exploratory movement through space[23]. AH encodes more global value information[15,36] and guides exploitation by means of an information-compressing strategy. In order to test whether the contribution of the hippocampus to reward learning varies along the long axis, we examined the contributions of PH and AH to exploration and exploitation in a continuous action space that requires mapping. We used the "clock task," where action values vary along an interval marked by time and visuospatial cues[37]. Participants need to explore this interval extensively to discover the most rewarding options[2]. Critically, on discrete-choice tasks (e.g., multiarmed bandits), it is hard to judge how exploratory a given choice is. In contrast, in continuous space we can define exploration in terms of the distance between consecutive choices, a spatial metric encoded in the human hippocampus[38], corresponding to trial-by-trial response time (RT) swings on the clock task.

To dissect the decision processes that underlie choices on this task, we applied our computational RL model (StrategiC Exploration/Exploitation of Temporal Instrumental Contingencies (SCEPTIC)), which learns the values of alternative actions using a basis function representation. Relative to traditional discrete-choice RL models, SCEPTIC provides a smooth approximation of the value function over the clock task interval (details below; Fig. 1). Thus, the model maps the global value maximum, allowing us to quantify its prominence as the reverse of Shannon's entropy (information content) of the value representation[35]. Our model can dissociate this global reinforcement statistic from state-wise RPEs. We observed a double dissociation wherein PH encodes local reinforcement (trial- and location-specific RPEs), whereas AH responds to the prominence of the global value maximum (low entropy). Furthermore, by comparing the neural fit of our information-compressing selective maintenance model to that of its full-maintenance counterpart, we demonstrate value information compression in AH. Consistent with functionally separable roles in resolving the explore/exploit dilemma, PH responses predicted further exploration, whereas AH responses predicted convergence on the global value maximum. Furthermore, AH displayed goal cell-like ramping responses as one approached the learned value maximum. Finally, responses to reinforcement were immediate and phasic in the PH, consistent with local processing and delayed and sustained in AH, consistent with integrative processing.

## Results

**Clock task: RT swings capture exploration.** On the clock task (Fig. 1a), participants explore and learn reward contingencies in a challenging unidimensional environment, namely a 4-s time interval. The passage of time is marked by the rotation of a dot around a clock face, reducing demands on internal timing. They were told to find the best RT based on reinforcement provided in the form of points. In each of the eight 50-trial blocks, one of the four contingencies with varying probability/magnitude tradeoffs determined the rewards. Two contingencies were learnable (increasing and decreasing expected value, IEV and DEV) and two were unlearnable (constant expected value, CEV, and constant expected value-reversed, CEVR, with a reversed probability-magnitude tradeoff). The task encourages extensive exploration and trial-by-trial learning. While people's responses shifted toward value maxima in learnable contingencies (Fig. 1c), even the more successful participants tended not to respond as early as possible in DEV. Likewise, most participants rarely responded as late as possible in IEV, where the value maximum has low probability. Thus, participants did not grasp that contingencies were monotonic, instead converging on a perceived value maximum in each block. Trial-wise changes of RTs (aka "RT swings") reflect the magnitude of exploration. Early in learning, better-performing participants displayed very high RT swings followed

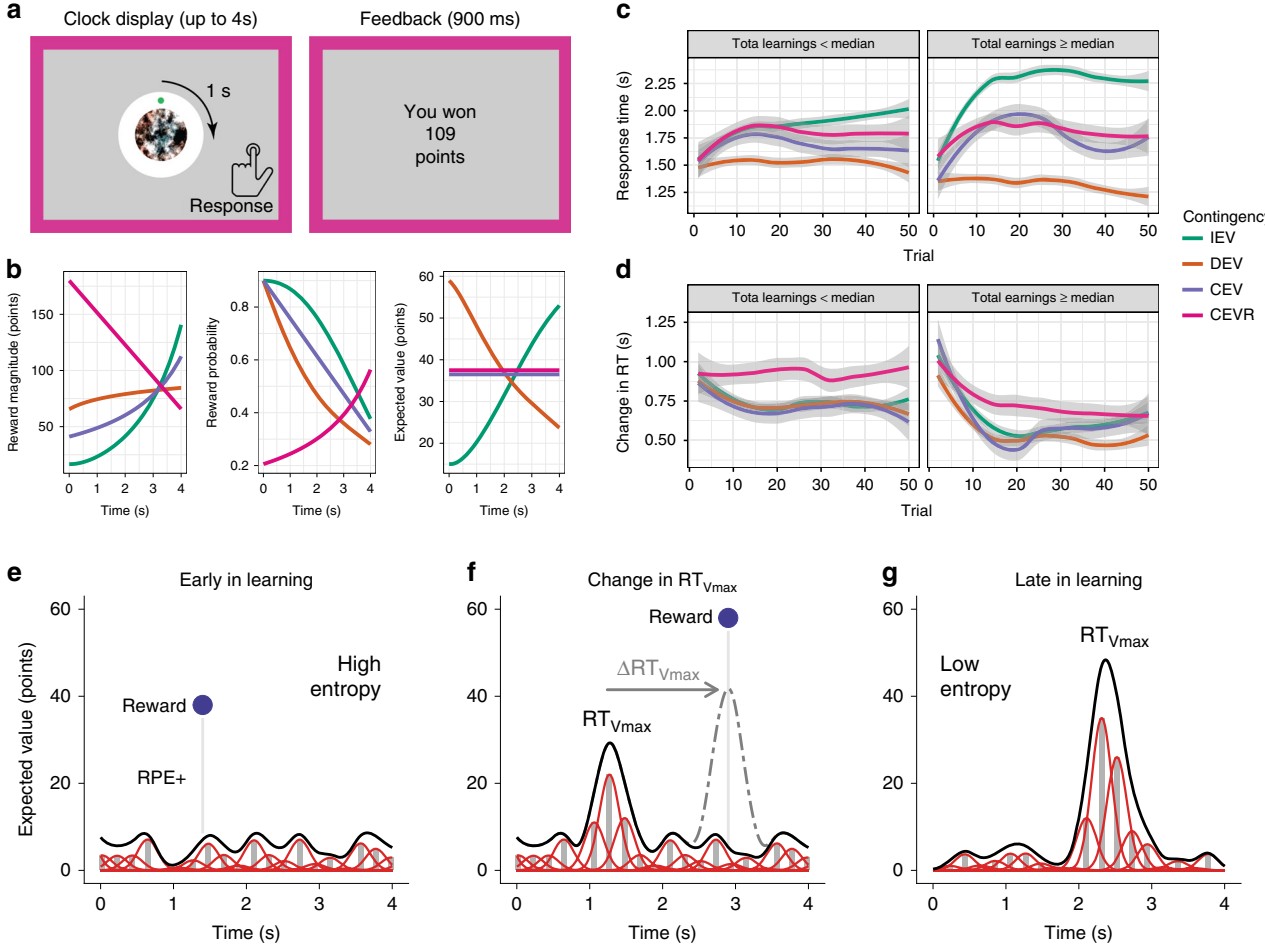

**Fig. 1 The clock paradigm, typical human behavior, and the SCEPTIC model. a** The clock paradigm consists of decision and feedback phases. During the decision phase, a dot revolves 360° around a central stimulus over the course of 4 s. Participants press a button to stop the revolution and receive a probabilistic outcome. **b** Rewards are drawn from one of four monotonically time-varying contingencies: increasing expected value (IEV), decreasing expected value (DEV), constant expected value (CEV), or constant expected value-reversed (CEVR). CEV and CEVR thus represent unlearnable contingencies with no true value maximum. Reward probabilities and magnitudes vary independently. **c** Evolution of subjects' response times (RT) by contingency and performance. Panels represent participants whose total earnings were above or below the sample median. Shaded error bands depict the standard error. $n = 70$ participants. **d** Evolution of subjects' response time swings (RT swings) by contingency and performance. $n = 70$ participants. **e** When all response times have similar expected values, the entropy of the value distribution is high, promoting entropy-guided exploration in the SCEPTIC model. A better-than-expected reward generates a positive reward prediction error (RPE+), which updates the value distribution. **f** Participants often respond near the response time of the global value maximum, $RT_{Vmax}$. However, on this trial the participant explores a later response time and receives a large unexpected reward, shifting the global value maximum, $\Delta RT_{Vmax}$, to a later time. **g** Late in learning, participants tend to converge on a perceived $RT_{Vmax}$ and to select response times near this "bump." Under the SCEPTIC model, values of preferred options are selectively maintained, whereas values of nonpreferred alternatives decay toward zero. The resulting value distribution has a prominent bump and lower entropy, promoting exploitative choices of high-value response times.

by a decline as they shift to exploiting the subjective value maximum. Less successful participants keep exploring stochastically, with moderately high RT swings throughout, and never settle on a clear value maximum. Curiously, successful participants transition from early exploration to later exploitation even in unlearnable contingencies where no objective value maximum exists, as we have reported previously (Fig. 1d[35]). As detailed in the next section, these results reflect how participants adaptively maintain value information.

**The SCEPTIC model captures local and global reinforcement.** Our SCEPTIC RL model[35] estimates local reinforcement (statewise RPEs) and global reinforcement (global value maximum). SCEPTIC approximates the expected value function along the time-varying reward contingency with a set of learning elements whose temporal receptive fields cover the 4-s trial interval[39,40].

Each element learns from temporally proximal rewards, updating its predicted reward (weight) by RPEs, which reflect the discrepancy between model-predicted reward at the chosen RT and the obtained reward (Fig. 1e). As detailed in "Methods," SCEPTIC learns the time-dependent contingency by integrating the delta learning rule[41] with a set of temporal basis functions (TBFs). The location of the global maximum (aka $RT_{Vmax}$) is defined as highest-valued RT within model-estimated value function (Fig. 1f). The prominence of the global value maximum relative to alternatives is quantified by Shannon's entropy of the normalized element weights, a log measure of the number of advantageous actions. Early in learning, the values of all actions are similar, entropy is high, and no clear global maximum exists. Later in learning, a subset of high-valued actions—or the global maximum—dominates, and the entropy declines. We have shown that selective maintenance of favored actions, compared to full

maintenance, accelerates the entropy decline later in learning, accentuating the global maximum, decreasing the amount of information held online, and facilitating the transition from exploration to exploitation[35].

**PH encodes prediction errors and AH, global value maximum.** We first examined neural encoding of local reinforcement in model-based whole-brain fMRI analyses. As expected, RPE signals were found in a canonical circuit encompassing the ventral striatum, thalamus, midbrain, and the cingulo-opercular (salience) network (see Supplementary Table 1). Activation in the bilateral PH was also detected at the whole-brain threshold (FWE-corrected $p < 0.05$, Fig. 2c, blue voxels). Responses to a prominent global value maximum (low entropy) were seen in the AH and the vmPFC (FWE-corrected $p < 0.05$; Fig. 2c, orange voxels; Supplementary Table 2).

Furthermore, a double dissociation emerged within the hippocampus, with PH selectively responding to RPEs and AH selectively responding to the global value maximum (Fig. 2b), anteroposterior location × signal $\chi^2(11) = 3235.36$, $p < 10^{-16}$. The posterior third of the hippocampus (four slices) was modulated by RPEs, adj. $ps < 0.01$, corrected for multiple comparisons using the method of Hothorn et al.[42] Conversely, the anterior two-thirds of the hippocampus (eight slices) was positively modulated by low entropy (quantified by the SCEPTIC selective maintenance model), adj. $ps < 0.01$.

One important question is whether RPEs in PH are indeed location-specific and do not simply signal changes in the overall reward rate. Supporting the former account, PH was more weakly modulated by RPEs from a standard delta rule learning model ($\alpha = 0.10$) that lacked a detailed representation of expected value across the interval (cf.[37]) compared to the SCEPTIC selective maintenance model, RPE type, $\chi^2(1) = 13.78$, $p < 0.0002$. SCEPTIC RPE modulation was particularly stronger than trial-level RPE modulation in the posterior quarter of the hippocampus (RPE type × anteroposterior location interaction $\chi^2(11) = 29.76$, $p < 0.001$, post hoc: adj. $ps < 0.01$ in posterior three slices). The superiority of SCEPTIC PE representations in the three most posterior hippocampal slices was qualitatively the same across a range of learning rates ($\alpha = 0.05–0.20$) for the standard delta rule model, all adj. $ps < 0.05$.

The SCEPTIC selective maintenance model further predicts that the mapping of the global value maximum depends on information compression whereby values of less preferred options are forgotten and preferred option values are selectively maintained (detailed in ref. [35]). Consistent with this prediction, AH responses to low entropy were only detected using estimates from the SCEPTIC selective maintenance model and not from its full-maintenance counterpart (Fig. 2d), anteroposterior location × SCEPTIC variant $\chi^2(11) = 187.27$, $p < 10^{-16}$. Entropy-related modulation was nonsignificant in all slices of the long axis according to the full-maintenance model (adj. $ps > 0.2$). These findings suggest that value representations in AH are compressed by selective maintenance.

**Hippocampal vs. other corticostriatal activation.** Given that we initially identified RPE and low-entropy activity using whole-brain analyses, we sought to examine whether individual differences in hippocampal responses to these signals were distinct from responses in other regions significant at the whole-brain level (Tables S1 and S2). More specifically, in exploratory factor analyses, we examined whether mean regression coefficients within the significant hippocampal clusters loaded onto the same latent factors as other corticostriatal coefficients. Individual differences in PH RPE responses loaded on a factor distinct from all other whole-brain-significant

RPE-sensitive regions (factor 1, 43% variance, encompassing the bilateral striatum, opercular-insular, and frontoparietal regions; factor 2, 20% variance, encompassing the bilateral PH; Supplementary Table 3). Analyses of entropy coefficients, however, revealed that low-entropy AH responses were on the same factor as vmPFC and ventral stream responses, suggesting shared representations (factor 1, 31% variance, encompassing high-entropy responsive dorsal attention network regions; factor 2, 28% variance, encompassing the left AH, vmPFC, fusiform gyrus, right operculum, and left precentral gyrus; Supplementary Table 4). Thus, for our analyses of behavioral relevance, we used PH RPE factor scores and the mean regression coefficient from the significant AH cluster as predictors.

**Posterior hippocampal prediction errors promote exploration.** If the PH binds states together, its activity should promote visits to remote states, or exploration. Indeed, individuals whose PH was more responsive to RPEs explored more, as indicated by larger RT swings (indicated by the effect of $RT_{t-1}$ on $RT_t$; $RT_{t-1} × PH$: $t = -11.31$, $p < 10^{-15}$, Fig. 2e; complete model statistics: Supplementary Table 5). Furthermore, these individuals were relatively more likely to make RT swings post reward, abandoning a just-rewarded location in favor of exploration ($RT_{t-1} ×$ last outcome × PH: $t = 5.71$, $p < 10^{-7}$). Confirming that these RT swings represented true exploration rather than a return to previously sampled high-value options, individuals with stronger PH RPEs chose lower-valued RTs following greater swings in learnable contingencies (RT swing × PH: $t = 2.28$, $p = 0.034$, RT swing × contingency × PH: $t = 2.73$, $p < 0.001$). Continual exploration on the clock task is predominantly stochastic, due to the difficulty of learning the values of the best RTs (high entropy); indeed, poorly performing participants exhibit persistently high RT swings (Fig. 1d). Interestingly, the effects of PH activity on exploration were not explained by poor task performance as reflected in high subject-level entropy or low subject-level maximum available value, ruling out the trivial explanation that people who responded randomly (e.g., from being off-task) experienced more surprising feedback, triggering PH responses (Supplementary Table 5). Furthermore, participants with stronger PH responses did not win fewer points in the learnable conditions (PH: $t = -0.18$, $p = 0.86$, PH × trial: $t = 0.05$, $p = 0.96$).

Participants completed two sessions of the clock task in counterbalanced order, one in the MR scanner and one during magnetoencephalography recording (MEG; only behavioral results are reported in this study). This allowed us to test whether hippocampal signals recorded with fMRI predicted behavior during the MEG session. The effect of PH RPE responses on RT swings replicated out of session ($RT_{t-1} × PH$: $t = -5.80$, $p < 10^{-8}$, $RT_{t-1} ×$ last outcome × PH: $t = 3.97$, $p < 10^{-4}$; Fig. 2e, Supplementary Table 6), suggesting that exploration-related PH responses did not merely encode visits to various states during the fMRI session, but reflected one's relatively stable tendency to explore.

**Anterior hippocampal value encoding promotes exploitation.** Stronger neural encoding of the global value maximum in the AH should promote exploitation. Indeed, people with the strongest AH responses to the global value maximum were more likely to choose RTs near it ($RT_{Vmax} × AH$: $t = 3.39$, $p < 0.001$). As expected, this convergence was strongest late in learning ($-1/$ trial $× RT_{Vmax} × AH$: $t = 2.86$, $p = 0.004$, Fig. 2h). AH responses had no significant effect on exploration ($RT_{t-1} × AH$: $t = 0.91$, $p = 0.36$, $RT_{t-1} ×$ last outcome × AH: $t = 1.85$, $p = 0.064$; Fig. 2f, Supplementary Table 5).

In the replication session, RTs in people with the strongest AH responses to the global value maximum in fMRI were also more likely to converge on the global maximum ($RT_{Vmax} × AH$: $t = 2.31$,

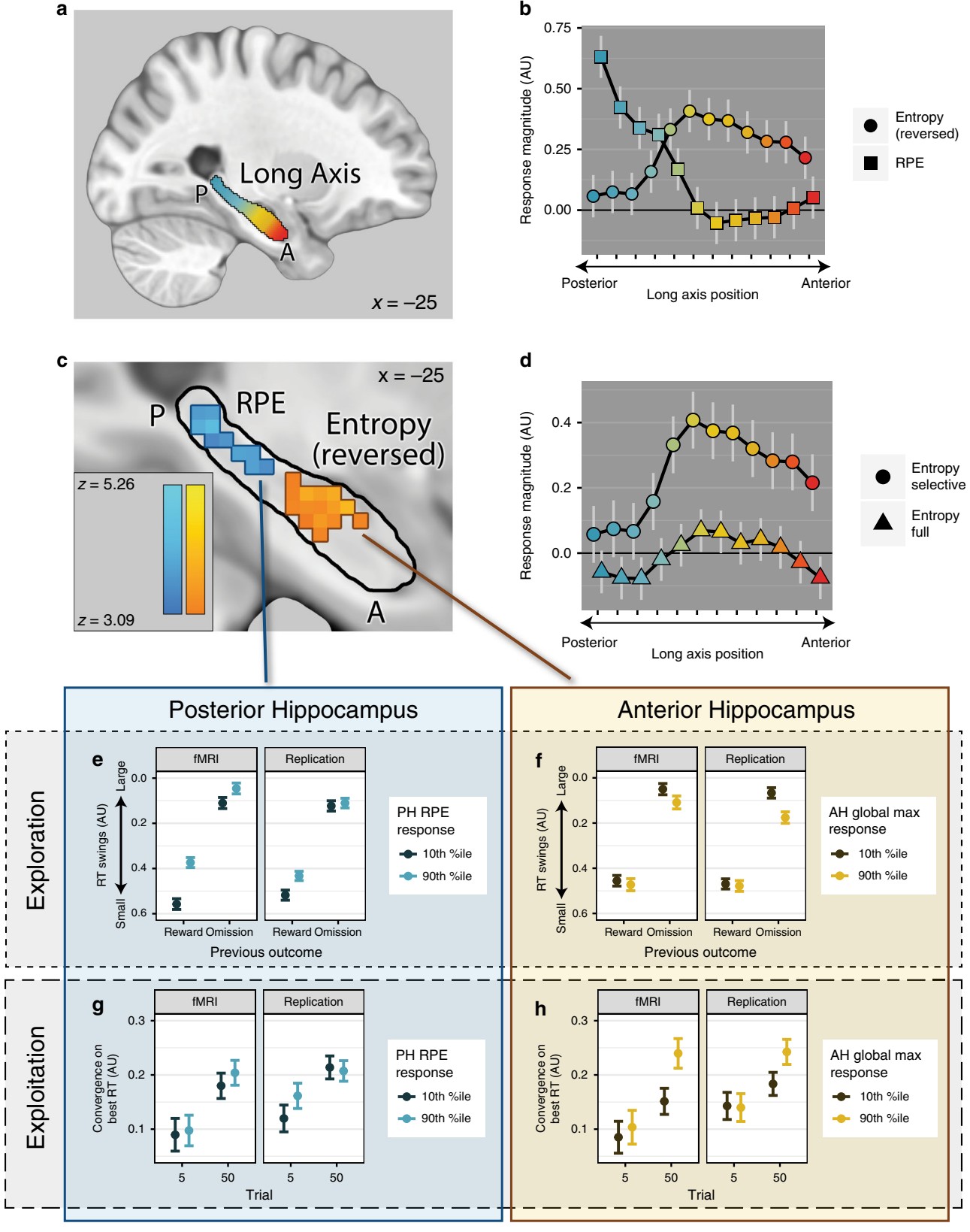

$p = 0.021$, Fig. 2h), particularly late in learning ($-1/\text{trial} \times \text{RT}_{\text{Vmax}} \times$ AH: $t = 3.11$, $p = 0.002$; details in Supplementary Table 6).

**PH/AH behavioral effects are not explained by confounds.** Critically, PH RPE and AH global value responses were not an artifact of novelty or some other time-dependent shift in activity unrelated to exploration/exploitation, as these signals persisted when early and late parts of each run were analyzed as separate regressors. More specifically, when we extracted general linear model (GLM) regression coefficients in the hippocampus from

**Fig. 2 Encoding of reinforcement along the A–P axis and its behavioral relevance. a** Long axis of the hippocampus. The coloration along the axis denotes the transition from more posterior (blue) to more anterior (red) portions. This color scheme is used throughout the paper to indicate how representation and effects on behavior vary along the long axis. The hippocampal mask is based on the Harvard-Oxford subcortical atlas. **b** Double dissociation of signals along the A–P axis: RPE responses predominate in PH and global value maximum responses, in AH. The light gray vertical lines denote the standard error and their centers denote the estimated mean from a multilevel regression model. $n = 70$ participants. **c** Prediction error responses in the PH and responses to low entropy (prominent global value maximum) in the AH, whole-brain FWE-corrected $p$s < 0.05. $n = 70$ participants. **d** The AH only tracks the prominence of the global value maximum as predicted by the information-compression selective maintenance SCEPTIC model, but not its full-maintenance counterpart. The light gray vertical lines denote the standard error and their centers denote the estimated mean from a multilevel regression model. $n = 70$ participants. **e–h** Double dissociation of behavioral correlates of PH vs. AH response. Full model statistics are presented in Supplementary Table 5. $n = 70$ participants, each tested in two independent sessions (fMRI and replication). Dots depict model-estimated means, error bars depict 95% CI. **e** PH RPE responses predict greater exploration, particularly after rewards. The ordinate axis in **e** and **f** denotes the autocorrelation between previous choice and the current choice, with higher values indicating greater RT swings. **f** AH global value maximum responses had no consistent relationship with exploration. **g** PH responses had no effect on exploitation. **h** AH responses predict greater exploitation, particularly late in learning. The ordinate axis in **g** and **h** denotes the effect of the $RT_{Vmax}$ on RT.

regressors representing the first and second halves of the task, the double dissociation between PH RPE and AH global value responses held in both the first half (trials 1–25; anteroposterior location × signal $\chi^2(11) = 1361.03$, $p < 10-16$) and second half (trials 26–50; anteroposterior location × signal $\chi^2(11) = 1685.54$, $p < 10-16$). In a model that treated run half as a categorical moderator, we found an anteroposterior location × signal × half interaction, $\chi^2(11) = 192.50$, $p < 10-16$, such that entropy modulation was more pronounced in mid-anterior slices early than late in learning, while RPEs became more focally associated with positive PH modulation late in learning (see Supplementary Fig. 2).

In further sensitivity analyses, we ascertained that the effects of PH vs. AH responses on exploration vs. exploitation were unchanged after controlling for behavioral variables (trial, contingency, maximum available value, uncertainty, and their interactions), subject-level performance (mean entropy and value) and for interactions between these potential confounds and hippocampal responses (Supplementary Table 5). Since we used the SCEPTIC RL model to generate RPE and entropy estimates, we further verified that our brain-behavior findings were not tautologically explained by inclusion of other model-derived covariates (e.g., maximum available value) into statistical models predicting behavior. Finally, given the established role of corticostriatal networks in reward learning, we ascertained that hippocampal signals predicted behavior above and beyond corticostriatal signals identified in the literature and our study (Tables S1–S4). Details of these analyses are provided in Supplementary Note 1.

**AH promotes while PH does not modify uncertainty aversion.** Our previous behavioral analysis of these data revealed that humans are uncertainty-averse in the large continuous space of the clock task, even after controlling for the value confound[35]. This was in part because they selectively remembered the values of preferred RTs, allowing the rest to decay, a form of information compression. At the same time, since PH responses were associated with exploratory RT swings, we sought to test whether they also predicted choosing relatively uncertain RTs. To test for uncertainty effects, we used a Kalman filter variant of SCEPTIC that estimated local uncertainty for each 0.1 s bin on each trial (see "Methods" for details). To test how hippocampal responses modified the influence of uncertainty, we predicted the hazard (i.e., momentary response probability conditional on not responding earlier) of making a response during the decision phase in a mixed effects continuous-time Cox survival model, treating uncertainty and value as time-varying covariates. This more nuanced analysis also accounts for censoring of later parts of the interval by earlier responses. Since individual SCEPTIC

model parameters may influence the scaling of value and uncertainty estimates[43], we rescaled value and uncertainty within participants to eliminate this confound. Our survival analyses confirmed that AH facilitated exploitation (AH × value: $z = 8.14$, $p < 10^{-15}$, see Supplementary Table 7 for full model statistics) and PH facilitated true exploration, i.e., a relative preference for lower-valued RTs (PH × $RT_{t-1}$: $z = 6.12$, $p < 10^{-9}$, PH × value: $z = -3.95$, $p < 0.0001$). The hypothesis of uncertainty-directed PH-driven exploration was not supported (PH × uncertainty: $z = -1.11$, $p = 0.27$). AH promoted uncertainty aversion (AH × uncertainty: $z = -2.60$, $p < 0.009$), supporting the hypothesis of AH information compression. Participants may miss the opportunity to respond early in the interval and also avoid the end of the interval to avoid forfeiting a reward for reasons unrelated to value or uncertainty. We censored these no-go zones, and the results remained qualitatively unchanged (AH × value: $z = 4.64$, $p < 10^{-5}$, PH × value: $z = -2.39$, $p = 0.017$, PH × uncertainty: $z = -0.26$, $p = 0.79$, AH × uncertainty: $z = -2.43$, $p < 0.015$). Confirming that these findings were not an artifact of predictor rescaling, the relevant effects remained and became stronger without the within-subject scaling of value and uncertainty ($|z| \geq 5.71$, $p < 10^{-7}$).

**Ramps in AH upon approach and departure from global maximum.** The preceding analyses show that AH session-level encoding of the global maximum location facilitates behavioral convergence on it (i.e., exploitation), but tell us little about real-time activity in the AH that may guide this convergence. Indeed, if AH represented the location of the global value maximum in a goal cell-like manner, we would expect its activity to increase on approach, as the most valuable action becomes available, and decrease when moving away. Whereas our model-based fMRI GLM analyses captured the average magnitude of responses in the AH across trials, they could not reveal the temporal dynamics of AH activity with respect to the global value maximum. To investigate these dynamics, we estimated real-time voxelwise hippocampal activity with a deconvolution algorithm[44], then event-locked the responses to the global value maximum (most advantageous RT in a given trial), resulting in a time course of activity for each trial. We examined these trial-wise hippocampal responses in multilevel models vis-à-vis behavioral variables (additional details in "Methods"). This analysis gives us a more direct view of hippocampal activity, overcoming the assumptions of the standard GLM, and it does not depend on predictions of the SCEPTIC model for decoding the BOLD signal.

In analyses of online responses (i.e., during the decision-making phase) activity in the AH but not PH ramped up toward the global maximum ($RT_{Vmax}$) and ramped down as the clock advanced past it (Fig. 3). We would expect such inverted-U ramps

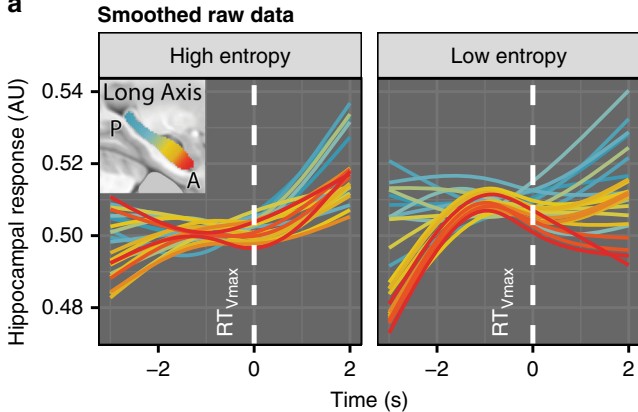

**a** Smoothed raw data

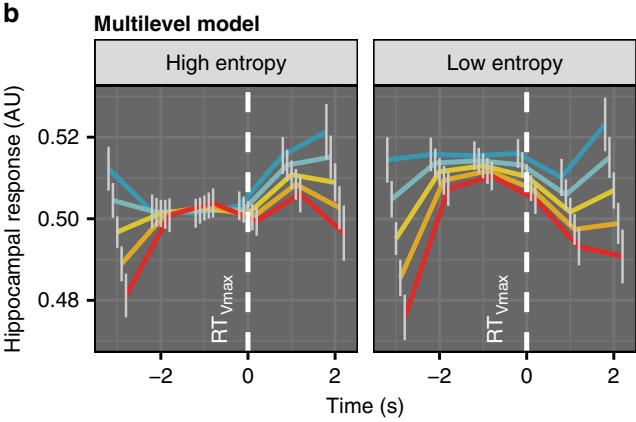

**b** Multilevel model

**Fig. 3 Online hippocampal responses time-locked to RT$_{Vmax}$, deconvolved BOLD signal. a** Raw data, GAM smoothing with three knots, voxelwise responses shown in 24 long-axis bins. $n = 70$ participants. RT$_{Vmax}$ is shown with the white dashed line). **b** Multilevel model, completely general time effect. Centers denote model-estimated mean, vertical gray lines denote standard errors from the multilevel model. $n = 70$ participants. NB: since voxelwise time series are normalized, only differences in the shape, but not intercept, of response can be interpreted.

to scale with the prominence of the global maximum relative to alternative RTs and this was the effect we observed. Specifically, inspection of smoothed raw data suggested the presence of inverted-U ramps aligned with the RT$_{Vmax}$ (Fig. 3a), particularly when entropy was low. A multilevel model with completely general time (i.e., treating time as an unordered factor to avoid parametric assumptions) revealed a time × anteroposterior location interaction ($\chi^2[5] = 43.8$, $p < 10^{-5}$) indicating more prominent ramps in AH than in PH (Fig. 3b), and a time × entropy interaction ($\chi^2[5] = 20.0$, $p = 0.001$), indicating more prominent ramps on low-entropy trials (the time × location × entropy interaction was not significant in this model). The activity in AH seemed highest one second before RT$_{Vmax}$, suggesting anticipation or response preparation. Furthermore, a more parsimonious model specifically testing linear and quadratic effects of time revealed a significant time$^2$ × location × entropy interaction ($\chi^2[1] = 5.4$, $p = 0.02$), in addition to the time$^2$ × anteroposterior location ($\chi^2[1] = 23.3$, $p < 10^{-5}$) and time$^2$ × entropy ($\chi^2[1] = 24.1$, $p < 10^{-6}$) interactions. This analysis suggested that activity ramps specifically in AH (vs. PH) were more prominent on low-entropy trials. Ramps were modulated by entropy, but not by preceding reward (time$^2$ × reward $\chi^2[1] = 0.1$, ns; time$^2$ × location × reward $\chi^2[1] = 0.01$, ns), indicating that they reflected global reinforcement aggregated

over multiple episodes rather than the immediately preceding episode.

**Responses to reinforcement in PH are early and phasic; delayed and sustained responses in AH encode the shifting global maximum.** Once the subject traverses the space obtaining a reward or omission, this reinforcement needs to be bound to the cognitive map, both across states (possible RTs) and learning episodes (trials). In order to examine how hippocampal activity during the post-feedback period may support integration of reinforcement into a structured representation, we aligned deconvolved hippocampal time series to the feedback period using the approach described above and detailed in "Methods." If PH bound recent rewards to local states, we would expect relatively early responses following feedback. Conversely, if AH integrated rewards across distant states and learning episodes, its responses might be later and slower. Indeed, PH exhibited rapid, on–off responses to reinforcement, whereas responses in AH were delayed and sustained (Fig. 4a, b). These differences were most pronounced after a reward (vs. omission, time point × anteroposterior location × reward: $\chi^2[9] = 23.7$, $p < 0.005$).

Time courses of post-feedback responses throughout learning also differed across the long axis. An analysis treating trial and location as completely general (i.e., unordered factors avoiding the parametric assumption that responses scale linearly with trial or location) revealed that AH responses increased more markedly than PH responses throughout the first 20 trials. As with responses to low entropy, this pattern was weaker in the anteriormost part of the head and in the PH (trial [five bins] × anteroposterior location [six bins]: $\chi^2[20] = 99.5$, $p < 10^{-11}$; Fig. 4c, d), suggesting greater integration of reinforcement across episodes in the anterior body.

Building on these descriptive results, we explored how the hippocampus encoded the location of the global value maximum (RT$_{Vmax}$) in the post-outcome time interval and across the long axis. To check the power of this analysis to detect the encoding of behavioral variables, we first examined the trivial effect of preceding trial's RT on voxelwise deconvolved signals, which was robust and positive throughout the long axis in the first 2 s after the outcome (Supplementary Fig. 3). Thus, our post-outcome analyses included preceding RT as a covariate to control for this possible confound. As an additional check, we reproduced our conventional whole-brain analysis finding of entropy signals in AH (Fig. 4e). Finally, our substantive analysis revealed that the RT$_{Vmax}$ location on the current trial was signaled throughout the long axis before and during feedback (Fig. 4f), while the shift in RT$_{Vmax}$ (cf. Fig. 1f) was signaled early in PH and then later and more prominently, in AH (Fig. 4g). Thus, when the global value maximum shifted closer (earlier in the interval) compared to the preceding trial due to reinforcement, AH activity increased.

## Discussion

Whereas previous studies of the explore/exploit dilemma have primarily focused on the neocortex, striatum and amygdala[2–6], we show that the hippocampus plays a key role in resolving this dilemma when values are organized spatially. Using a basis function RL model of a unidimensional continuous space, we observed doubly dissociated representations of reinforcement along the hippocampal long axis: rapidly evolving state-wise RPE signals in the PH facilitated exploration and slowly evolving global value maximum signals in the AH drove the transition to exploitation.

We found that RPEs in the human PH invigorated exploration, as shown by greater distances between consecutive choices, shifts toward lower-valued options and costly win-shift responses.

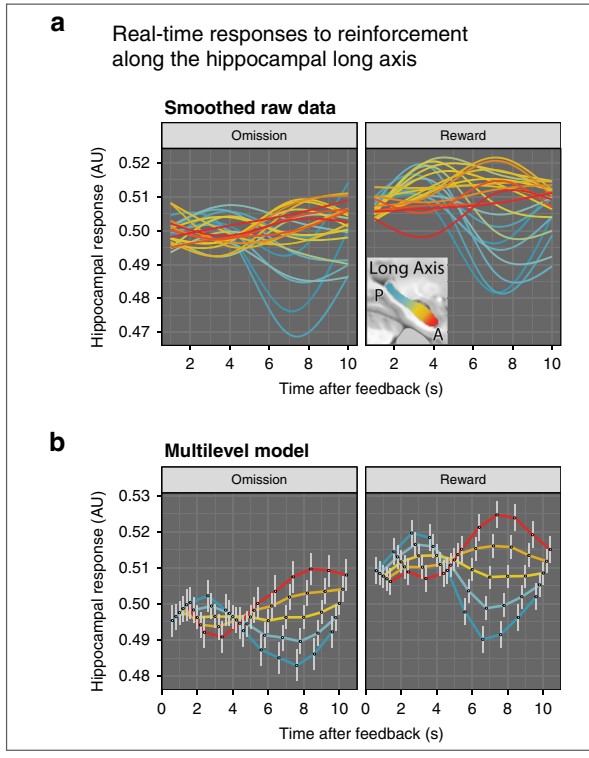

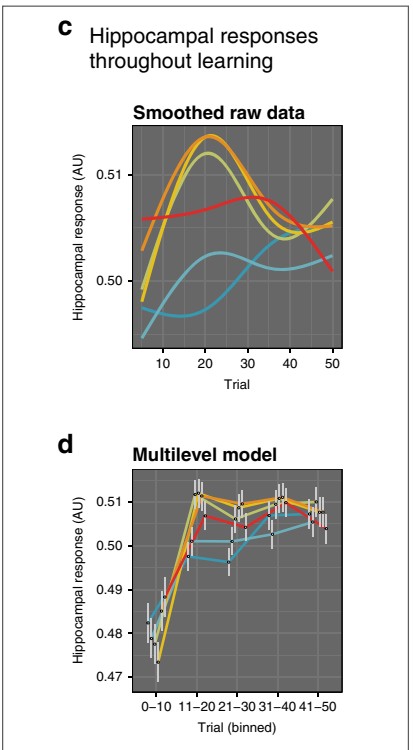

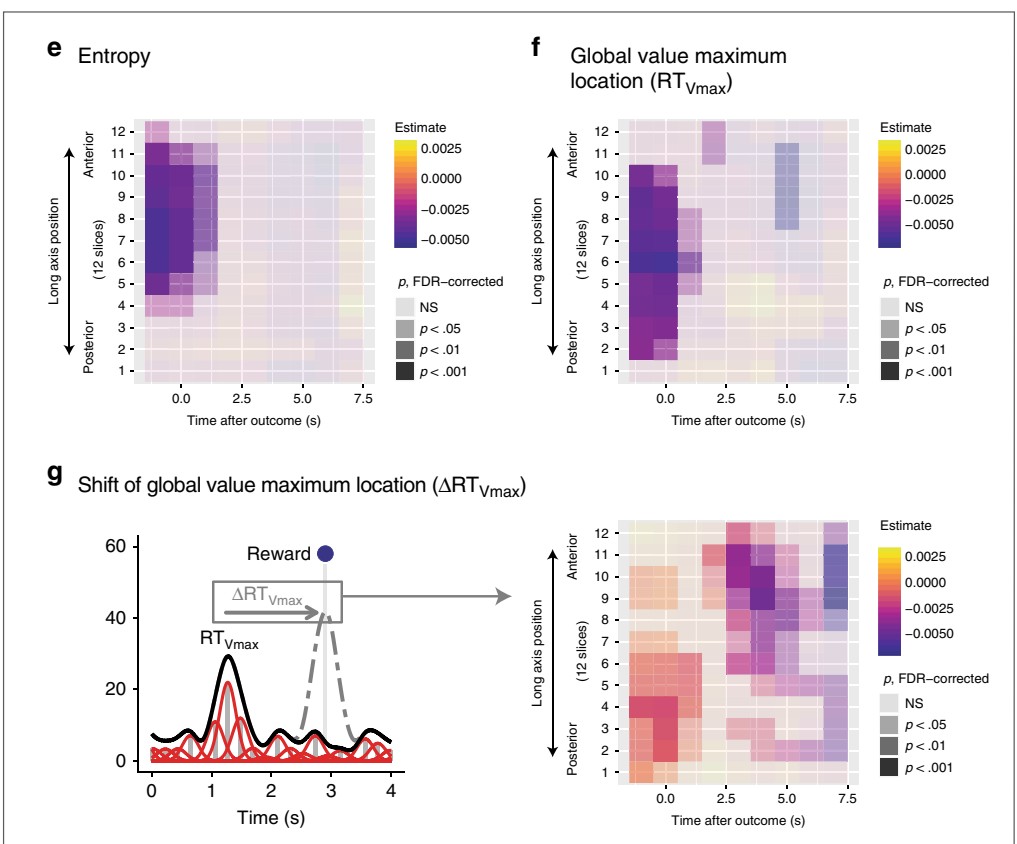

These exploratory shifts were not driven by uncertainty: participants, regardless of the strength of their PH RPE responses, avoided more uncertain parts of the interval. PH may thus simply drive random exploration, akin to increasing softmax temperature. It is also possible that PH invigorates a systematic movement through space unguided by value or uncertainty. PH-mediated exploration is consistent with the finding that optogenetic stimulation of the rodent dorsal dentate gyrus (DG) granule cells promotes exploration of novel environments[23]. Notably, dorsal DG-mediated exploration depends on dopaminergic input[23], supporting the idea that exploration is invigorated by dopaminergic RPE signals. Indeed, RPEs are found in rodent dorsal hippocampus[24] and may depend on the dopaminergic inputs from the VTA[21] and the LC[22], which enhance memories

**Fig. 4 Real-time responses to reinforcement along the hippocampal long axis. a** Responses during the ITI time-locked to feedback, raw data with GAM smoothing, three knots. **b** Responses during the ITI time-locked to feedback, multilevel general linear model with completely general time and bin location (12 bins) as predictors. Centers denote model-estimated mean, vertical gray lines denote standard errors from the multilevel model. $n = 70$ participants. **c** Evolution of hippocampal responses to reinforcement across trials, raw data with GAM smoothing, three knots. $n = 70$ participants. **d** Evolution of hippocampal responses to reinforcement across trials, multilevel general linear model with completely general learning epoch (five 10-trial bins) and bin location (12 bins) as predictors. $n = 70$ participants. **e** Unfolding hippocampal responses to prior entropy (before current reinforcement) time-locked to current reinforcement, multilevel model. Negative regression coefficients indicate stronger responses to low entropy (prominent global value maximum). **f** Unfolding hippocampal responses to prior global value maximum location (before current reinforcement) time-locked to current reinforcement, multilevel model. Negative regression coefficients indicate stronger responses to a more proximal (earlier) global value maximum. **g** Unfolding hippocampal responses to prior the shift in the global value maximum location following current reinforcement time-locked to current reinforcement, multilevel model. Negative regression coefficients indicate stronger responses when the global value maximum moves closer (earlier in the interval).

for novel events[45], and on functional connections with reward-sensitive ventral striatal neurons[46]. While previous imaging studies using spatially unstructured paradigms have generally not detected RPE signals in human PH (Supplementary Fig. 1)[27,28], some reported analogous deactivations to error[47] and activations to reward[48].

By contrast, the human AH tracked the global value maximum, both across episodes as participants' choices converged on it (Figs. 2h and 4d–g), and within-episode as they navigated toward or away from the best RT (Fig. 3). Our SCEPTIC model predicts that values of nonpreferred actions are compressed out late in successful learning, accentuating the global maximum and promoting exploitation[35]. Indeed, only the information-compressing model and not its otherwise identical counterpart predicted AH responses (Fig. 2d), indicating that a fine-grained representation of values in the environment is compressed to summary statistics of a single global maximum or "value bump." Furthermore, stronger AH responses predicted avoidance of uncertain options beyond the degree predicted by the SCEPTIC model, pointing to additional mechanisms through which AH shifts the choices toward the rich parts of the environment, away from unrewarding or uncertain alternatives. The functional co-activation of the vmPFC and AH to low entropy suggests that their interactions[29,49–51] may facilitate the binding of compressed value representations into a map that guides choices toward preferred options[34,52–54] and shifts between egocentric and allocentric navigation[34].

During online navigation, AH responses ramped up in anticipation of the global value maximum in a manner reminiscent of dopamine ramps in the mesostriatal circuit[55,56]. After the space was traversed and the outcome was obtained, PH displayed early, on–off reward-modulated responses. AH reward-modulated responses, on the other hand, were delayed and sustained. Furthermore, AH encoded directional shifts of the global value maximum (Fig. 4g). The coupling between hippocampal BOLD response and theta power is poorly understood[57–59]. Nevertheless, the delay between PH and AH in both overall responses to reinforcement (Fig. 4a,b) and specifically in responses to the advancing global maximum (Fig. 4g) matches the posteroanterior (in rodents: dorsoventral) direction of traveling theta waves[60,61]. Our observations are thus consistent with a unidirectional spread of information from PH to AH, with AH integrating reinforcement across states and episodes. These responses could also correspond to diverging replay patterns in the PH vs. AH, with PH replaying actual and counterfactual trajectories toward recently obtained rewards and AH replaying trajectories leading to value maxima in a goal cell-like pattern[7,62,63].

Responses within the AH were heterogenous: sustained post-reinforcement signals were strongest in the anterior body, whereas online goal cell-like responses to the global value maximum were most evident in the head. Aside from long-axis

location, this may reflect the folding of human AH, with the anterior portion of the head being comprised mostly of CA3/CA1 and lacking the DG[64]. Thus, goal cell-like responses in the head likely originate in the CA3/CA1 or the subiculum and not in the DG. More speculatively, it is possible that the global maximum is encoded in the early nodes of the trisynaptic pathway (DG) and that its location and prominence are signaled in the hippocampal output from CA1 during online navigation[65].

Among the strengths of our study are the task and an information-compressing basis function RL model that give us access to a spatially structured value vector, dissociating the global value maximum from local RPEs. Hippocampal representations of these signals were not simply explained by novelty or epoch in learning. This approach echoes earlier models of hippocampal learning with a basis function representation of continuous states or actions[63,66]. Our results could not have been obtained with a spatially unstructured paradigm. Our novel multilevel analysis of deconvolved BOLD signals revealed the within-trial temporal dynamics of hippocampal responses and allowed us to detect goal cell-like activity. This approach may prove useful for testing strong hypotheses about functional gradients in regional activation, especially for event-related designs where trials are sampled at multiple TRs and jittered intertrial interval (ITIs) offer a window into poststimulus processing. State-of-the-art fMRI methods including high spatial ($2.3\text{ mm}^3$) and temporal ($\text{TR} = 1\text{ s}$) resolution, a large number of trials ($n = 400$), and a reasonably large sample ($n = 70$) allowed us to detect hippocampal reward signals generally not observed in earlier studies (Supplementary Fig. 1). Finally, out-of-session replication of brain-behavior relationships strengthens the case for hippocampal contributions to exploration and exploitation.

Within the inherent constraints of fMRI, our design and analyses provide excellent resolution on coordinated neural activity, yet these constraints also preclude us from addressing questions about cell-level representations and oscillations in the hippocampus. Furthermore, our whole-brain analyses revealed distributed responses to both RPEs and entropy across frontostriatal circuits. Yet we did not further investigate the interactions between the hippocampus and regions such as vmPFC, which may be important for anticipating upcoming rewards[67]. Our experiment provided reinforcement based on response timing; thus, participants always traversed the environment in a single direction. Future experiments with $k$-dimensional spaces could, for example, test for goal-like AH responses more robustly by controlling participants' movement relative to the goal and, in general, establish whether our findings generalize beyond the time domain. It also remains unclear whether our findings generalize to environments with rewards distributed less smoothly and even discontinuously. Such Easter egg environments in which reward-rich locations are hidden among reward-poor areas are harder to capture by a coarse representation, making value information less compressible. Our computational experiments using the

SCEPTIC model, however, show that even value functions containing discontinuous local maxima can be effectively compressed using a policy that selectively maintains preferred options[35]. Finally, while our sample varied substantially in age (14–30), we did not test for age-related changes in the hippocampus to exploration or exploitation. This is an important topic for future research given emerging evidence of changes in both exploration[68,69] and learning from ambiguous and aversive outcomes[70,71] between adolescence and adulthood.

Altogether, our findings revealed that PH and AH exert complementary influences on value-guided choices, with PH invigorating exploration that updates local values and AH promoting exploitative choices of the action perceived to be the best. Combined, these processes use reinforcement to guide allocentric navigation and stand in contrast to egocentric win-stay/lose-shift responses supported by the amygdala[72] or learning of spatially unstructured values in the mesostriatal circuit[73].

## Methods

**Participants**. Participants were 70 typically developing adolescents and young adults aged 14–30 ($M = 21.4$, SD = 5.1). Thirty-seven (52.8%) participants were female and thirty three were male. Prior to enrollment, participants were interviewed to verify that they had no history of neurological disorder, brain injury, pervasive developmental disorder, or psychiatric disorder (in self or first-degree relatives). Participants and/or their legal guardians provided informed consent or assent prior to participation in this study. Experimental procedures for this study complied with Code of Ethics of the World Medical Association (1964 Declaration of Helsinki) and the Institutional Review Board at the University of Pittsburgh (protocol PRO10090478). Participants were compensated $75 for completing the experiment.

**Behavioral task**. Participants completed eight runs of the exploration and learning task (aka the clock task, based on Moustafa et al.[37]) during an fMRI scan, implemented in MATLAB 2012a and PsychToolbox 3.0.10. Runs consisted of 50 trials in which a green dot revolved 360° around a central stimulus over the course of 4 s (see Fig. 1a). Participants pressed a button to stop the dot, which ended the trial. They then received a probabilistic reward for the chosen RT according to one of four time-varying contingencies, two learnable (increasing and decreasing expected value, IEV and DEV) and two unlearnable. All contingencies were monotonic but featured reward probability/magnitude tradeoffs that made learning difficult. RT swings were the index of exploration (Badre et al.[2]). After each response, participants saw the probabilistic reward feedback for 0.9 s. If participants failed to response within 4 s, they received 0 points. Each trial was followed by an ITI that varied in length according to an exponential distribution. To maximize fMRI detection power, the sequence and distribution ITIs were derived using a Monte Carlo approach implemented by the *optseq2* command in *FreeSurfer* 5.3. More specifically, we simulated five million possible ITI sequences consisting of 50 trials each and retained the top 320 orders based on their estimation efficiency. For each subject, the experiment software randomly sampled 8 of these efficient ITI sequences, which were used for the durations of ITIs in the task.

The central stimulus was a face with a happy expression or fearful expression, or a phase- scrambled version of face images intended to produce an abstract visual stimulus with equal luminance and coloration. Faces were selected from the NimStim database[74]. All four contingencies were collected with scrambled images, whereas only IEV and DEV were also collected with happy and fearful faces. The effects of the emotion manipulation will be reported in a separate manuscript because they are not central for the examination of the neural substrates of exploration and exploitation on this task. Likewise, age-related differences in brain activity will be reported separately. We note that fitted parameters for the SCEPTIC model did not vary significantly as a function of age ($ps > 0.2$), though overall performance (number of points earned) increased somewhat with age, $r = 29$, $p = 0.01$.

As part of a larger study, participants also completed this task during a separate magnetoencephalography (MEG) session. The order of the fMRI and MEG sessions was counterbalanced (fMRI first $n = 34$, MEG first $n = 36$) and the sessions were separated by 3.71 weeks on average (SD = 1.59 weeks). The behavioral data from the MEG session were used for out-of-session replication tests in which we examined how brain activity during the fMRI scan predicted behavior during the MEG session. (Neural data for the MEG study will be reported separately.) This enabled us to establish whether individual differences in hippocampal activity and exploration/exploitation represented stable tendencies vs. patterns incidental to a single experimental session.

**Neuroimaging acquisition**. Neuroimaging data during the clock task were acquired in a Siemens Tim Trio 3T scanner at the Magnetic Resonance Research Center, University of Pittsburgh. Due to the varying RTs produced by participants as they learned the task, each fMRI run varied in length from 3.15 to 5.87 min ($M = 4.57$ min, SD = 0.52). Imaging data for each run were acquired using a simultaneous multislice sequence sensitive to BOLD contrast, TR = 1.0 s, TE = 30 ms, flip angle = 55°, multiband acceleration factor = 5, voxel size = 2.3mm³. We also obtained a sagittal MPRAGE T1 scan, voxel size = 1mm³, TR = 2.2 s, TE = 3.58 ms, GRAPPA 2x acceleration. The anatomical scan was used for coregistration and nonlinear transformation to functional and stereotaxic templates. We also acquired gradient echo fieldmap images (TEs = 4.93 and 7.39 ms) for each subject to quantify and mitigate inhomogeneity of the magnetic field across the brain.

**Preprocessing of neuroimaging data**. Anatomical scans were registered to the MNI152 template[75] using both affine (ANTS SyN) and nonlinear (FSL FNIRT) transformations. Functional images were preprocessed using tools from NiPy[76], AFNI (version 19.0.26)[77], and the FMRIB software library (FSL version 6.0.1)[78]. First, slice timing and motion coregistration were performed simultaneously using a four-dimensional registration algorithm implemented in NiPy[79]. Nonbrain voxels were removed from functional images by masking voxels with low intensity and by a brain extraction algorithm implemented in the program *ROBEX*[80]. We reduced distortion due to susceptibility artifacts using fieldmap correction implemented in FSL FUGUE.

The participants' functional images were aligned to their anatomical scan using the white matter segmentation of each image and a boundary-based registration algorithm[81], augmented by fieldmap unwarping coefficients. Given the low contrast between gray and white matter in echoplanar scans with fast repetition times, we first aligned functional scans to a single-band fMRI reference image with better contrast. The reference image was acquired using the same scanning parameters, but without multiband acceleration. Functional scans were then warped into MNI152 template space (2.3 mm resolution) in one step using the concatenation of functional-reference, fieldmap unwarping, reference-structural, and structural-MNI152 transforms. Images were spatially smoothed using a 5 mm full-width at half maximum (FWHM) kernel using a nonlinear smoother implemented in FSL SUSAN. Whereas all voxels were spatially smoothed in our whole-brain analyses, our detailed analyses of hippocampal time courses used a 5 mm FWHM smoother within the anatomical mask to reduce partial volume effects (details below). To reduce head motion artifacts, we then conducted an independent component analysis for each run using FSL MELODIC. The spatiotemporal components were then passed to a classification algorithm, ICA-AROMA, validated to identify and remove motion-related artifacts[82]. Components identified as noise were regressed out of the data using FSL regfilt (nonaggressive regression approach). ICA-AROMA has performed very well in head-to-head comparisons of alternative strategies for reducing head motion artifacts[83]. We then applied a 0.008 Hz temporal high-pass filter to remove slow-frequency signal changes[84]; the same filter was applied to all regressors in GLM analyses. Finally, we renormalized each voxel time series to have a mean of 100 to provide similar scaling of voxelwise regression coefficients across runs and participants.

**Computational modeling of behavior: general approach**. Behavior was fitted with the SCEPTIC RL model[35]. Building on models of Pavlovian conditioning of Ludvig et al.[39], SCEPTIC uses Gaussian TBFs to approximate the time-varying instrumental contingency. Each function has a temporal receptive field with a mean and variance defining its point of maximal sensitivity and the range of times to which it is sensitive. The weights of each TBF are updated according to a delta learning rule. While in temporal difference (TD) models, learning and choice take place on a moment-to-moment basis, humans tend to strategically consider the decision space as a whole[37]. Accordingly, SCEPTIC applies updates and makes choices at the trial level. Crucially, SCEPTIC maintains action values selectively, allowing for forgetting of action values not selected on the current trial. Selective maintenance facilitates the transition from exploration to exploitation in computational experiments and accounts for uncertainty aversion in humans[35].

Model parameters were fitted to individual choices using an empirical Bayesian version of the variational Bayesian approach[85]. The empirical Bayes approach relied on a mixed effects model in which individual-level parameters are assumed to be sampled from a normally distributed population. The group's summary statistics, in turn, are inferred from individual-level posterior parameter estimates using an iterative variational Bayesian algorithm in which the algorithm alternates between estimating the population parameters and the individual subject parameters. Over algorithm iterations, individual-level priors are shrunk toward the inferred parent population distribution, as in standard multilevel regression. Furthermore, to reduce the possibility that individual differences in voxelwise estimates from model-based fMRI analyses reflected differences in the scaling of SCEPTIC parameters, we refit the SCEPTIC model to participant data at the group mean parameter values. This approach supports comparisons of regression coefficients across subjects and also reduces the confounding of brain-behavior analyses by the individual fits of the computational model to a participant's behavior. We note, however, that our fMRI results were qualitatively the same when model parameters were free to vary across people (additional details available from the corresponding author upon request).

To examine the emergence of the global value maximum that guides the transition from initial exploration to exploitation, we estimated Shannon's entropy

(or information content) of the normalized vector of TBF weights (action values). Entropy provides a log measure of the number of good actions (in this case, temporal segments). Entropy is high during the initial exploration, when action values are close and decreases as one action begins to dominate, corresponding to the perceived global value maximum. These entropy dynamics are only observed under selective maintenance, which compresses the amount of information retained later in learning and accentuates the global value maximum[35].

**Core architecture of SCEPTIC model**. The SCEPTIC model represents time using a set of unnormalized Gaussian radial basis functions (RBFs) spaced evenly over an interval $T$ in which each function has a temporal receptive field with a mean and variance defining its point of maximal sensitivity and the range of times to which it is sensitive, respectively (a conceptual depiction of the model is provided in Fig. 1). The primary quantity tracked by the basis is the expected value of a given choice (RT). To represent time-varying value, the heights of each basis function are scaled according to a set of $b$ weights, $\mathbf{w} = [w_1, w_2,...,w_b]$. The contribution of each basis function to the integrated value representation at the chosen RT, $t$, depends on its temporal receptive field

$$\varphi_b(t) = \exp\left[-\frac{(t-\mu_b)^2}{2s_b^2}\right], \tag{1}$$

where $\mu_b$ is the center (mean) of the RBF and $s_b^2$ is its variance. And more generally, the temporally varying expected value function on a trial $i$ is obtained by the multiplication of the weights with the basis

$$V(i) = \mathbf{w}(i)\varphi. \tag{2}$$

In order to represent decision-making during the clock task, where the probability and magnitude of rewards varied over the course of 4-s trials, we spaced the centers of 24 Gaussian RBFs evenly across the discrete interval and chose a fixed width, $s_b^2$, to represent the temporal variance (width) of each basis function. More specifically, $s_b^2$ was chosen such that the distribution of adjacent RBFs overlapped by approximately 50% (for additional details and consideration of alternatives, see ref. [35]).

The model updates the learned values of different RTs by updating each basis function $b$ according to the equation

$$w_b(i+1) = w_b(i) + e_b(i|t)\alpha[\text{reward}(i|t) - w_b(i)], \tag{3}$$

where $i$ is the current trial in the task, $t$ is the observed RT, and reward $(i|t)$ is the reinforcement obtained on trial $i$ given the choice $t$. The effect of prediction error is scaled according to the learning rate $\alpha$ and the temporal generalization function $e_b$. To avoid tracking separate value estimates for each possible moment, it is crucial that feedback obtained at a given RT $t$ be propagated to adjacent times. Thus, to represent temporal generalization of expected value updates, we used a Gaussian RBF centered on the RT $t$, having width $s_g^2$ and normalized to have an area under the curve of unity. The eligibility of a basis function $\varphi_b$ to be updated by prediction error is defined by the area under the curve of its product with the temporal generalization function

$$e_b(i|t) = \int_0^T \mathcal{N}(t, s_g^2)\varphi_b dt. \tag{4}$$

This parameterization leads to a scalar value for each RBF between 0 and 1 representing the proportion of overlap between the temporal generalization function and the receptive field of the RBF. In the case of perfect overlap, where the RT is perfectly centered on a given basis function and the width of the generalization function matches the basis (i.e., $s_g^2 = s_b^2$), $e_b$ will reach unity, resulting a maximal weight update according to the learning rule above. Conversely, if there is no overlap between an RBF and the temporal generalization function $e_b$ will be 0 and no learning will occur in the receptive field of that RBF.

The SCEPTIC model selects an action based on a softmax choice rule, analogous to simpler RL problems (e.g., two-armed bandit tasks[1]). For computational speed, we arbitrarily discretized the interval into 100 ms time bins such that the agent selected among 40 potential responses. The agent chose responses in proportion to their expected value

$$p(rt(i+1) = j|V(i)) = \frac{\exp(V(i)_j/\beta)}{\sum_{t=0}^T \exp(V(i)_t/\beta)}, \tag{5}$$

where $j$ is a specific RT and the temperature parameter, $\beta$, controls the sharpness of the decision function (at higher values, actions become more similar in selection probability).

Importantly, as described extensively in our earlier behavioral and computational paper[35], a model that selectively maintained frequently chosen high-value actions far outperformed alternative models. More specifically, in the selective maintenance model, basis weights revert toward 0 in inverse proportion to the temporal generalization function:

$$w_b(i+1) = w_b(i) + e_b(i|t)\alpha[\text{reward}(i|t) - w_b(i)] - \gamma(1 - e_b(i|t))(w_b(i) - h), \tag{6}$$

where $\gamma$ is a selective maintenance parameter between 0 and 1 that scales the degree

of reversion toward a point $h$, which is taken to be 0 here, but could be replaced with an alternative, such as a prior expectation. As detailed in our previous report, late in learning, selective maintenance compresses the amount of value information represented by the agent by 1/3 to 1/2 (more in exploitative subjects) and accelerates the transition from exploration to exploitation by accentuating the global value maximum and effacing the values of nonpreferred segments[35]. All of our primary fMRI analyses were based on signals derived from fitting the selective maintenance SCEPTIC model to participants' behavior.

As noted in the "Results" section, we sought to examine whether anterior hippocampal responses to low entropy were specific to the selective maintenance model, consistent with information compression. To test the specificity, we compared entropy representation from the SCEPTIC selective maintenance mode to a full-maintenance counterpart that did not decay the values of the unchosen RTs (more detailed model comparisons provided in ref. [35]). More specifically, the learning rule for the full-maintenance model was

$$w_b(i+1) = w_b(i) + e_b(i|t)\alpha[\text{reward}(i|t) - w_b(i)]. \tag{7}$$

**Quantification of uncertainty**. In our earlier computational modeling and behavioral analyses of these data[35], we tested a number of alternative models, including those that explicitly represented sampling uncertainty about alternative actions. More specifically, these models implemented variants of a Kalman filter for each TBF such that the basis approximated both the posterior expectation (i.e., mean) and uncertainty (i.e., standard deviation) for each possible RT. Although uncertainty-tracking models were inferior in behavioral Bayesian model comparisons, for our neural analyses, we nevertheless wished to examine whether the hippocampus may be involved in promoting or discouraging actions based on their uncertainty.

Therefore, we estimated a Kalman filter variant (hereafter called Fixed U+V) in which a fixed learning rate was used for updating the expected value, whereas the posterior uncertainty estimates were updated according to the Kalman gain. The learning rule for Fixed U+V was

$$\mu_b(i+1) = \mu_b(i) + e_b(i|t)\alpha[\text{reward}(i|t) - \mu_b(i)], \tag{8}$$

where $\mu_b(i)$ represents the expected value of basis function $b$ on trial $i$, and $\alpha$ represents the learning rate. The gain for a given basis function, $k_b(i)$ is defined as

$$k_b(i) = \frac{\sigma_b(i)^2}{\sigma_b(i)^2 + \sigma_{\text{rew}}^2}, \tag{9}$$

where $\sigma_{\text{rew}}^2$ represents the expected volatility (measurement noise) of the environment. Here, we provide the model the variance of returns from a typical run of the experiment as an initial estimate of measurement noise, although other priors lead to similar model performance. We also initialize prior estimates of uncertainty for each basis function to be equal to the measurement noise, $\sigma_{b0}^2 = \sigma_{\text{rew}}^2$, leading to a gain of 0.5 on the first trial (as in ref. [86]).

Under the KF, uncertainty about expected value for each basis function is represented as the standard deviation of its Gaussian distribution. Likewise, posterior estimates of uncertainty about responses proximate to the basis function $b$ decay in inverse proportion to the gain according to the following update rule

$$\sigma_b(i+1) = [1 - e_b(i|t)k_b(i)]\sigma_b(i). \tag{10}$$

Estimates of the time-varying value and uncertainty functions are provided by the evaluation of the basis over time

$$V(i) = \mu(i)\varphi. \tag{11}$$

$$U(i) = \sigma(i)\varphi. \tag{12}$$

The Fixed U+V policy represents a decision function, $Q(i)$, as a weighted sum of the value and uncertainty functions according to a free parameter, $\tau$. As uncertainty decreases with sampling and expected value increases with learning, value-related information will begin to dominate over uncertainty. Positive values of $\tau$ promote uncertainty-directed exploration, whereas negative values yield uncertainty aversion

$$Q(i) = V(i) + \tau U(i). \tag{13}$$

For the purpose of fMRI analysis, we fit the Fixed U+V model to participants' behavior, then extracted trial-wise estimates of uncertainty. More specifically, we obtained the model-estimated uncertainty of the chosen action for each trial. Given that uncertainty for a given process decays exponentially under a KF approach, we computed the percentile of the uncertainty of the chosen action relative to the alternative actions on the same trial. This trial-wise normalization ensured that the fMRI analyses of uncertainty were not confounded by slower changes in overall uncertainty over the entire learning episode.

**Trial-level alternative model of RL**. The SCEPTIC model is based on a TBF architecture that provides a state-wise representation of value and RPEs (i.e., the model estimates these quantities at every RT within each trial). A simpler alternative is that participants represent value and RPEs at the whole-trial level, instead tracking the expected value of responding during the trial and not discriminating among alternative RTs. This alternative model was considered primarily to test

whether posterior hippocampal RPE responses were more consistent with SCEPTIC state-wise RPEs or simpler trial-level RPEs. More specifically, the alternative model was a variant of the Rescorla–Wagner delta rule

$$V(i+1) = V(i) + \alpha[\text{reward}(i) - V(i)], \tag{14}$$

where $i$ denotes the trial and $\alpha$ is the learning rate. For simplicity, we tested the performance of this model using learning rates in the set, $\alpha = \{0.05, 0.1, 0.15, 0.2\}$.

**Conceptual comparison of SCEPTIC model to earlier time clock (TC) model**. Previous papers describing behavior on the clock task have suggested that some humans tend to shift toward more uncertain RTs[86] and that this tendency is associated with greater activity in the rostrolateral prefrontal cortex[2]. These findings are largely founded on a different computational model of the task, called the TC model, which represents RTs on each trial $i$ as a linear combination of several potentially neurobiological processes

$$\widehat{\text{RT}}(i) = K + \lambda \text{RT}(i-1) + \nu\left[\text{RT}_{\text{best}} - \text{RT}_{\text{avg}}\right] - \text{Go}(i) + \text{NoGo}(i) \\ + \rho\left[\mu_{\text{slow}}(i) - \mu_{\text{fast}}(i)\right] + \varepsilon\left[\sigma_{\text{slow}}(i) + \sigma_{\text{fast}}(i)\right]. \tag{15}$$

The details of each parameter and the underlying representation are provided in previous reports[86]. Briefly, however, with respect to value-based decisions, the TC model separately updates the probability of a positive prediction error for RTs that are slower or faster than the subject's average ($\mu_{\text{slow}}$ and $\mu_{\text{fast}}$, respectively). With learning, the model predicts that subjects shift toward faster or slower RTs that are associated with a greater expectation of positive prediction errors according to a free parameter, $\rho$. The definitions of "fast" and "slow" responses are based on a comparison to the running average of recent RTs. TC tracks the expected value ($\mu$) and uncertainty ($\sigma$) using two beta distributions, one for "fast" and one for "slow" responses. Our previous computational and behavioral analyses found that the TC model has problems with parameter identifiability, that its substantive parameters for value and uncertainty do not contribute to model fit in empirical data, and that the model performs poorly in more complex time-dependent contingencies[35].

Perhaps more important than these limitations are the conceptual differences between the TC and SCEPTIC models, which render SCEPTIC particularly well suited for detailed analyses of exploration and exploitation on the clock task. The representation of value over time involves a tradeoff between the generality of representation on one hand and the number of free parameters or values stored on the other. A completely general temporal value representation is exemplified by TD models, which we have previously tested. On the other end, parsimonious parametric models such as Frank's TC[37] often can only explain a narrow range of phenomena; they break down more easily at boundary conditions.

RBF representation, in our opinion, finds the middle ground between these two extremes: it reduces the memory and computational load compared to TD, while maintaining generality of representation, which enables it to learn virtually any contingency in one continuous dimension. Furthermore, by approximating the value function over the time interval of the task, the SCEPTIC model enables one to test hypotheses about both the chosen action (e.g., its expected value, or RPE) and global statistics such as the entropy of the value function. Moreover, the function approximation approach of SCEPTIC can be extended to test whether humans prefer or are averse to more uncertain options (the Fixed U+V model above). Thus, variants of the SCEPTIC model can disentangle stochastic vs. uncertainty-related exploration on the clock task. The former is related to the entropy of learned values that enter into the softmax choice rule; the latter depends on explicit tracking of the sampling uncertainty in a Kalman filter. By comparison, the fast vs. slow parametric representation of TC provides a coarser view of the task that does not distinguish between stochastic and uncertainty-directed exploration and or provide the statistics of the global value maximum.

**Voxelwise general linear model analyses**. Voxelwise GLM analyses of fMRI data were performed using FSL version 6.0.1[78]. Single-run analyses were conducted using FSL FEAT v6.0, which implements an enhanced version of the GLM that corrects for temporal autocorrelation by prewhitening voxelwise time series and regressors in the design matrix[84]. For each design effect, we convolved a duration-modulated unit-height boxcar regressor with a canonical double-gamma hemodynamic response function (HRF) to yield the model-predicted BOLD response. All models included convolved regressors for the clock and feedback phases of the task.

Moreover, GLM analyses included parametric regressors derived from SCEPTIC. For each whole-brain analysis, we added a single model-based regressor from SCEPTIC alongside the clock and feedback regressors. Results were qualitatively unchanged, however, when all SCEPTIC signals were included as simultaneous predictors, given the relatively low correlation among these signals. We further verified that the key double dissociation between prediction errors and entropy along the long axis of the hippocampus (Fig. 2) held when entropy and RPEs were included simultaneously in the fMRI GLMs. As shown in Supplementary Fig. 4, there was no meaningful difference in the double dissociation when GLM coefficients were extracted from models with one model-based regressor each (i.e., separate models for entropy and prediction errors) vs. a model that included both of these regressors simultaneously.

Importantly, the results of these and other fMRI analyses would only diverge if the model-based regressors had a moderate to strong correlation with each other, leading to collinearity problems. To examine this possibility, we computed the correlation between the convolved regressors for RPEs and entropy for all subjects and runs. We then modeled the correlation in a Bayesian multilevel model (implemented in the *brms R* package[87]) that included a random intercept of subject and allowed for heterogeneity between runs in the variability of the RPE-entropy correlation. This analysis revealed a very small average correlation between PE and entropy, $r = 0.07$, 95% highest posterior density interval = 0.05–0.09. Following Cohen's rules of thumb, we further tested for the probability that the RPE-entropy correlation is small, $|r| < 0.10$ using a region of parameter equivalence test on the posteriors from the Bayesian multilevel model. This test revealed that 100% of the posterior samples of the PE-entropy correlation fell within this range, providing strong evidence that the correlation between entropy and PE is small. Altogether, the low level of correlation between these convolved model-based signals indicates that any additional analyses based on regression coefficients from the fMRI GLMs would be very similar regardless of whether the signals were modeled individually or simultaneously, consistent with Supplementary Fig. 4.

For each model-based regressor, the SCEPTIC-derived signal was mean-centered prior to convolution with the HRF. The RPE signal was aligned with the feedback, whereas entropy and uncertainty were aligned with the clock (decision) phase. Furthermore, for regressors aligned with the clock phase, which varied in duration, we sought to unconfound the height of the predicted BOLD response due to decision time from the parametric influence of the SCEPTIC signal. Toward this end, for each trial, we convolved a duration-modulated boxcar with the HRF, renormalized the peak to 1.0, multiplied the regressor by the SCEPTIC signal on that trial, then summed across trials to derive a single model-based regressor (cf. processing time vs. intensity of activation in (cf. processing time vs. intensity of activation in ref. [88]). This approach is equivalent to the dmUBLOCK(1) parameterization provided by AFNI for duration-modulated regressors in GLM analyses.

Parameter estimates from each run were combined using a weighted fixed effects model in FEAT that propagated error variances from the individual runs. The contrasts from the second-level analyses were then analyzed at the group level using a mixed effects approach implemented in FSL FLAME. Specifically, we used the FLAME 1+2 approach with automatic outlier deweighting[89], which implements Bayesian mixed effects estimation of the group parameter estimates including full Markov Chain Monte Carlo-based estimation for near-threshold voxels[90]. In order to identify statistical parametric maps that best represented the average response, all group analyses included age and sex as covariates of no interest (esp. given the developmental sample).

To correct for familywise error at the whole-brain level, we computed the voxelwise residuals of a one-sample $t$-test for each contrast of interest in the group analysis, then generated 10,000 null datasets by randomizing the sign of the residuals (implemented by AFNI *3dttest++ -Clustsim*). These null datasets were then analyzed to identify the threshold for clusters that were significant at a whole-brain level at $p < 0.05$ (implemented by AFNI *3dClustsim*). For these calculations, we used a voxelwise threshold of $p < 0.001$[91]. Importantly, the sign randomization approach does not assume any parametric form for the spatial autocorrelation of the data, overcoming concerns about high false positive rates for cluster thresholding methods that assume a Gaussian autocorrelation function[92]. Cluster thresholds were 107 voxels for RPE analyses and 117 voxels for entropy analyses.

In addition to mitigating head motion-related artifacts using ICA-AROMA, we excluded runs in which more than 10% of volumes had a framewise displacement (FD) of 0.9 mm or greater, as well as runs in which head movement exceeded 5 mm at any point in the acquisition. This led to the exclusion of 11 runs total, yielding 549 total usable runs across participants. Furthermore, in voxelwise GLMs, we included the mean time series from deep cerebral white matter and the ventricles, as well as first derivatives of these signals, as confound regressors[83].

**Analyses of hippocampal responses**. We used a hippocampal parcellation from the Harvard-Oxford subcortical atlas to define bilateral masks for the hippocampus in the MNI152 space. The atlas was resampled to 2.3 mm voxels to match the functional data, then thresholded at 0.5 probability, yielding masks of 393 voxels in the left hemisphere and 401 voxels in the right hemisphere. To define the long axis, we identified the ten most anteroinferior and posterosuperior voxels in each hemisphere mask. We then took the centroid of these voxels and computed the slope of a regression line that connected these coordinates. We averaged the slopes for the left and right hemispheres to compute the optimal rotation of the coordinate space along the long axis of the hippocampus. We computed the slope difference of this average line relative to the anterior commissure-posterior commissure (AC–PC) axis, which has a zero slope in the sagittal plane. This yielded a rotation of 42.9° clockwise relative to the AC–PC axis. Finally, we verified this transformation by eye (gradient depicted in Fig. 2a).

While we view the inclusive Harvard-Oxford mask as more appropriate given the spatial smoothness of BOLD data and coregistration noise, in supplementary analyses, we also considered a more restrictive hippocampal mask derived using a detailed anatomical segmentation approach developed by Winterburn et al.[93]. Briefly, this segmentation approach was applied to the original anatomical scans forming the MNI152 template set, yielding a parcellation already in the

MNI152 space (publicly available here: https://github.com/CoBrALab/atlases/tree/master/mni_models/nifti). We retained the following regions from the parcellation in the mask: CA1, CA4/DG, CA2/CA3, subiculum, and stratum. These masks (265 voxels in the left hippocampus, 273 in the right) were approximately one third smaller than the Harvard-Oxford masks. The results using were qualitatively the same regardless of the mask (see Supplementary information for details; Supplementary Figs. 5, 6).

To examine how individual differences in hippocampal responses along the long axis relate to behavior, we extracted regression coefficients (aka "betas") from model-based whole-brain fMRI GLM analyses. We first extracted betas from clusters surviving whole-brain thresholding. For each signal—entropy, expected value, RPE —clusters were subjected to between-subject exploratory factor analysis (principal axis factoring with oblimin oblique rotation) to identify separable components representing each signal. We evaluated the number of factors based on very simply solution and Velicer's minimum average partial criteria[94]. These analyses were largely motivated to examine whether hippocampal responses were separable from other corticostriatal regions.

To relate hippocampal betas to exploratory and exploitative choices on the task, we regressed trial-wise RTs on trial-level signals such as previous outcome, $RT_{Vmax}$, and previous RT, as well as subject-level signals, particularly betas from the posterior and anterior hippocampal clusters identified in whole-brain analyses. Testing cross-level interactions, we examined how hippocampal responses moderated the effects of behavioral variables, such as the tendency to explore or convergence on $RT_{Vmax}$. We fitted multilevel regression models using restricted maximum likelihood estimation in the *lme4* package[95] in $R$[96], allowing for a random intercept of subject and run nested within subject.

Building on our whole-brain voxelwise analyses, we examined representations of decision signals along the hippocampal long axis. To support these analyses, we extracted voxelwise $z$-statistics within the hippocampal mask for RPEs, entropy of the value distribution, and relative uncertainty of the chosen action. We note that using normalized betas in these analyses yielded identical results; we preferred $z$-statistics because they better accommodate run variation in the precision of effects within the GLM framework. To analyze $z$-statistics along the long axis, we binned voxelwise statistics into 12 quantiles of even size (i.e., approximately equal numbers of voxels per bin) along the long axis. Aggregating the voxels of each bin, we computed the mean $z$ statistic for relevant decision signals and analyzed responses to entropy and RPEs along the long axis (Fig. 2b, d).

Although betas from fMRI GLMs provide a useful window into how decision signals from SCEPTIC relate to behavior at the level of an entire session, the GLM approach makes a number of assumptions: (1) that one correctly specifies when in time a signal derived from a computational model modulates neural activity, (2) that there is a linear relationship between the model signal and BOLD activity, and (3) that a canonical HRF describes the BOLD activity corresponding to a given model-based signal. Furthermore, a conventional model-based fMRI GLM does not allow one to interrogate whether the representation of a given cognitive process varies in time over the course of a trial. For these reasons, we conducted additional analyses that could provide a detailed view of how hippocampal activity changes both during and following each trial on the clock task. These analyses also attempted to overcome statistical and conceptual limitations of the GLM and to provide an index of within-trial neural activity that was independent of our computational model.

We first applied a leading hemodynamic deconvolution algorithm to estimate neural activity from BOLD data[44]. This algorithm has performed better than alternatives in simulated and real fMRI data, and it is reasonably robust to variations in the timing of neural events and the sampling frequency of the scan[97]. Within our anatomical mask of the bilateral hippocampus, we deconvolved the BOLD activity for each voxel time series and retained these as a voxels × time matrix for each run of fMRI data. In addition, to reduce the possibility that activity estimates reflected the influence of voxels outside of the hippocampus, for deconvolution, we used fMRI data in which spatial smoothing was applied only within the anatomical mask. More specifically, we applied a 5 mm FWHM smoothing kernel within the hippocampal mask using the AFNI *3dBlurInMask* program. The fMRI data for deconvolution analyses were otherwise preprocessed using the same pipeline described above.

Then, to estimate hippocampal activity for each trial in the experiment, we extracted the deconvolved signal in two epochs: (1) online (clock onset to RT) responses time-locked to $RT_{Vmax}$, (±3 s) censoring feedback and ITI periods, and (2) feedback onset and ITI (−1 to +10 s; the second preceding feedback was included for reference). This windowing approach allowed us to examine hippocampal activity during online decision-making in the clock task, as well as offline activity during the ITI. Given the fast event-related design, however, the onset of the next trial in the experiment may have occurred before 10 s post feedback had elapsed. In these cases, trial-wise estimates of post-feedback activity were treated as missing for all times after the onset of the next trial. The exponential distribution of ITI times yielded more data for activity proximate to the onset of feedback, but there were still several trials per subject with ITIs of 10 s or greater. Finally, to ensure that discrete-time models of neural activity could be easily applied, we resampled deconvolved neural activity onto an evenly spaced 1 s grid aligned to the event of interest using linear interpolation. The sampling

frequency of the fMRI scan was also 1 s. Thus, this interpolation was a form of resampling, but did not upsample or downsample the data in the time domain.

To link real-time hippocampal responses with behavior and decision signals from the SCEPTIC model, we divided hippocampal voxels into 12 even bins along the long axis, mirroring the regression beta analyses described above (illustrations of smoothed raw data use 24 bins for within-trial time courses and six bins for across trials time courses to aid readability). For each trial and time point within trial, we averaged voxels within each long axis bin. For each subject, this yielded a 400 trial × 11 time point (0–10 s) × 12 bin matrix for the feedback-aligned data. We then concatenated these matrices across participants for group analysis. Within each time × bin combination, we regressed trial-wise neural activity on key decision variables in a multilevel regression framework implemented in *lmer* in R, allowing for crossed random intercepts of subject and side (right/left).

To examine the temporal dynamics of hippocampal reinforcement representations in greater detail, we considered treating both time and bin as unordered factors in a combined multilevel regression model, rather than running separate models by time and bin. Although statistically estimable, these models were unwieldy because of the number of higher-order interactions. Instead, to adjust for multiple comparisons in nonindependent models separately examining each time point and bin, we applied the Benjamini–Yekutieli correction across models to maintain a false discovery rate of 0.05.

**Analyses of behavior using frequentist multilevel models**. Since our behavioral observations had a clustered structure (e.g., trials nested within subjects), we used multilevel regression models to estimate the effects of interest. Multilevel models were estimated using restricted maximum likelihood in the *lme4* package[95] in R 3.4.0[96]. Estimated $p$ values for predictors in the model were computed using Wald chi-square tests and degrees of freedom were based on the Kenward–Roger approximation. Most multilevel regressions were run on trial-level data in order to capture the temporal dynamics of learning and performance. To test temporal precedence in trial-level data (e.g., previous reward predicting a change in current RT swing), relevant predictors were lagged by one trial. For trial-level analyses, subject and run were treated as random effects. In particular, many models examined whether a given decision signal from the SCEPTIC model moderated the influence of previous choice ($RT_{t-1}$) on current choice ($RT_t$) or RT autocorrelation. A weaker autocorrelation indicates greater RT swings, and variables that decrease autocorrelation are considered to increase exploration. While the absolute RT difference between consecutive trials used in earlier studies[2] seems to be an intuitive metric of RT swings, it suffers from several measurement problems. First, it has an inherently zero-inflated distribution and cannot be treated as approximately normally distributed in statistical models. Second, due to time-varying imprecision, this absolute difference scales with the RTs. Third, it depends on where the preceding RT is relative to the edge of the interval. Thus, the effect of $RT_{t-1}$ on $RT_t$ provides a more precise and less biased estimate of RT swings.

We also performed survival analyses predicting the temporal occurrence of response. These mixed effects Cox models (R coxme package)[98] aimed to examine the effects of model-predicted expected value and uncertainty on the likelihood of response, and the impact of session-level hippocampal responses on value- and uncertainty-sensitivity. This survival analysis does not assume that the subject pre-commits to a given RT, instead modeling the within-trial response hazard function in real, continuous time[99]. The survival approach accounts for censoring of later within-trial time points by early responses. Most importantly, it assumes a completely general baseline hazard function, allowed to vary randomly across participants. We thus avoid assumptions about the statistical distribution of RTs and account for trial-invariant influences such as urgency, processing speed constraints or opportunity cost. We also modeled only the 1000–3500 ms interval, excluding early RTs that may be shorter than the deliberation and motor planning period and the end of the interval which one may avoid in order to not miss responding on a trial. We included learned value from the selective maintenance model and uncertainty from the Kalman filter uncertainty + value model as time-varying covariates, sampled every 100 ms. Subject-specific intercept was included as a random effect.

**Reporting summary**. Further information on research design is available in the Nature Research Reporting Summary linked to this article.

## Data availability

All core datasets reported in this paper are publicly available here: https://doi.org/10.5281/zenodo.3978642.

## Code availability

The codes for all analyses reported in this study are available at: https://zenodo.org/badge/latestdoi/22355039.

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

## Acknowledgements
This work was funded by K01 MH097091, R01 MH067924, and R01MH10095 from the National Institute of Mental Health. The authors thank Jiazhou Chen (data processing) and Kai Hwang and Rajpreet Chahal (data collection). The authors also thank Vishnu Murty and Brad Wyble for helpful comments on an earlier draft of the manuscript.

## Author contributions
Conceptualization: M.N.H., B.L., and A.Y.D. Software: M.N.H. and A.Y.D. Formal analysis: A.Y.D. and M.N.H. Investigation: M.N.H. and B.L. Resources: M.N.H. and B.L. Data curation: M.N.H. Writing—original draft: A.Y.D. and M.N.H. Writing—review and editing: M.N.H., B.L., and A.Y.D. Project administration: M.N.H. and B.L. Funding acquisition: B.L., M.N.H., and A.Y.D.

## Competing interests
The authors declare no competing interests.
