## [Peer Review File · Nature Communications]

Reviewers' comments:

Reviewer #1 (Remarks to the Author):

In this study, the authors showed the functional relationship between activity along the long axis of the hippocampus and explore-exploit behavior. To do this, the authors had participants complete Michael Frank's "clock" task while in the scanner. In this task, participants must learn what response time is the most rewarding in a given condition, but can only do so by trying different response times (exploring) and learning from reinforcement to exploit a value maximum. They then fit this behavior to their previously published SCEPTIC model, which allows them to dissociate between exploration-based local reinforcement and exploitation-based reward maximization through the mechanism of information compression. They find that activity in the posterior hippocampus relates to exploratory behavior in the task while the anterior hippocampal activity more in line with value-driven exploitation. They also show that this activation patterns in both the anterior and posterior hippocampus differ in terms of time-scale, lending further evidence to a functional dissociation of these two regions during exploration and exploitation.

This paper is an important contribution to both the hippocampal literature and the literature on explore-exploit decision making. As far as the reviewers are aware no work in humans has connected these two literatures empirically in this way. It builds on previous work in rodents that finds the anterior and posterior hippocampus have different, but complementary roles in goal-directed behavior. It applies this to exploration and exploitation by a careful and thorough analysis of both behavior and activation, showing that theory-supported evidence of information compression may occur across the hippocampal axis and this supports the transition from exploratory to exploitative states.

In addition, the manuscript is clearly written, the study well powered (including a behavioral replication), and the data and code are available. We believe it is appropriate for publication in Nature Communications.

We do have some concerns that we would like to see addressed:

- * Comparison between findings in this task and previous studies. Michael Frank has published several papers on this task including at least one fMRI paper. How do the whole brain results compare with previously published findings? Of course, one problem here is the different model used by Frank et al. which brings us to a second point

- * Could the authors briefly discuss differences with and presumably advantages of their model compared to the original model? This may be most appropriate in the methods section, but to readers familiar with the explore-exploit literature, the SCEPTIC model is still new and the benefits for studying exploration in this task would be worth pointing out.

- * There are some papers that should probably be cited to connect this with the broader explore-exploit literature. Most notably the work by Wu and Schulz who have a few papers on spatially structured explore-exploit tasks with spatial correlations between bandits determined by a Gaussian process. This I believe is the first

Wu, C.M., Schulz, E., Speekenbrink, M. et al. Generalization guides human exploration in vast decision spaces. *Nat Hum Behav* 2, 915–924 (2018). <https://doi.org/10.1038/s41562-018-0467-4>

In addition, modern views of explore-exploit decision making focus on two types of exploration – directed (information seeking) and random (behavioral variability). This view was synthesized by Wilson, R. C., Geana, A., White, J. M., Ludvig, E. A., & Cohen, J. D. (2014). Humans use directed and random exploration to solve the explore–exploit dilemma. *Journal of Experimental Psychology: General*, 143(6), 2074.

But nicely summarized in this review paper by Schulz and Gershman

Schulz, E., & Gershman, S. J. (2019). The algorithmic architecture of exploration in the human brain. *Current opinion in neurobiology*, 55, 7-14.

In this regard, it would be nice to have a paragraph in the discussion connecting the findings to directed and/or random exploration.

- * The sample for this study includes a wide age range (from 14 to 30). Previous work on exploration has shown that there are differences between adolescence and young adults in terms

of their exploratory behavior, particularly with regards to directed (or exploration driven by information seeking)

Schulz, E., Wu, C. M., Ruggeri, A., & Meder, B. (2019). Searching for rewards like a child means less generalization and more directed exploration. *Psychological science*, 30(11), 1561-1572.

Somerville, L. H., Sasse, S. F., Garrad, M. C., Drysdale, A. T., Abi Akar, N., Insel, C., & Wilson, R. C. (2017). Charting the expansion of strategic exploratory behavior during adolescence. *Journal of experimental psychology: general*, 146(2), 155.

And also differences in ambiguity aversion between teens and young adults

Tymula, A., Belmaker, L. A. R., Roy, A. K., Ruderman, L., Manson, K., Glimcher, P. W., & Levy, I. (2012). Adolescents' risk-taking behavior is driven by tolerance to ambiguity. *Proceedings of the National Academy of Sciences*, 109(42), 17135-17140.

Do you find age differences in terms of behavior within the task? And do patterns of task-related activation differ by age? Regardless of the result, a short paragraph connecting to this previous literature would be good.

Reviewer #2 (Remarks to the Author):

I was very happy to receive the manuscript by Dombrovski and colleagues. The study adds to a growing literature of hippocampal contributions to reinforcement learning, here, specifically exploring the neural mechanisms that underly the arbitration between exploration/exploitation in task that require cognitive mapping of task space. Although there is an extensive literature on how the cerebral cortex and basal ganglia may contribute to this computation, little attention has been directed at the hippocampus, which, based on its computational properties, is well suited to play a role in this tradeoff.

The theoretical justifications for the study are sound, the paradigm is well established, the computational and imaging methods appear rigorous and appropriate, the study is well powered (including out of session replication). The results, I believe, will also be of great interest to the wider field. In addition I find the writing clear.

Of particular interest is the described double dissociation along the long axis within the hippocampus, through a number of model-based imaging analyses (based on a computational RL model which uses a set of temporally extended basis functions to represent the task space), tying the posterior part to exploration and the anterior part to exploitation.

I have a few requests:

1. Can the authors please define 'RT swing' more clearly. For being such an important regressor I was surprised that I could not find exactly how it was computed. Is it simply the absolute RT difference between RT on trial N and trial N-1?

2. In the section 'Voxelwise general linear model analyses' the authors describe that only one model-based regressor was added at a time in the parametric fMRI models. It is reassuring that the results are allegedly quantitatively similar when all SCEPTIC regressors were included, however, since these analyses provide the basis for the main results it would be good to see this elaborated on in the paper with additional analysis (perhaps supplemental) rather than simply upon request. I do not know what "relatively low correlation" means in this context and I would like to be reassured that the main results survives taking all the correlated regressors into account.

3. As far as I understand the learning rate is a free parameter in the SCEPTIC model, however, the trial-level alternative model of reinforcement learning, learning rate α is set to .1. Does the supremacy of SCEPTIC hold across the whole range of learning rates in the simpler model?

4. I found it very interesting that PH RPE responses loaded on a factor distinct from all other whole-brain-significant RPE-sensitive regions, while low-entropy AH responses were on the same factor as ventromedial prefrontal cortex. What does this mean for the double dissociation and how should this be interpreted in terms of neural mechanisms supported by the hippocampus and other brain regions? possible to say anything about mechanism given this result? Could it imply that PH RPE responses are simply inherited from other brain regions while there is some representation of value of states in AH that are computed there? I wouldn't mind some additional elaboration on the implications of this result.

On a related note, did the authors you ever attempt causal modelling to try and disambiguate whether the described regressors are computed in the hippocampus or elsewhere (I'm not saying that causal modelling is necessary but asking out of interest)?

Reviewer #3 (Remarks to the Author):

In this manuscript, the authors examine hippocampal responses on a reinforcement learning task, with a particular interest in understanding the explore-exploit trade off. Using a clever behavioral task, they had participants learn different response strategies suitable for different distributions of expected value across strategy space. They report a double dissociation of hippocampal activity, with posterior hippocampus encoding moment by moment reward prediction errors as subjects learn the best response strategy, and anterior hippocampus encoding a representation of value over a longer time horizon.

I liked the way the authors frame this task as a continuous, spatially-structured complement to bandit tasks with discrete options. The goal of extending the reinforcement learning framework from experimentally-tractable but less realistic situations to problems mimicking the complex spatial and temporal dependencies in the real world make this work timely and interesting, in my opinion. Overall, I found the paper well written, if a bit difficult to parse in some areas. However, I think the analyses and results are compelling, and are a valuable contribution to the literature implicating hippocampus in RL processes.

Below are some suggestions for improving the manuscript from its current form.

1) More info about the model.

As a general comment, I'd suggest adding more explanatory information about the task and especially the SCEPTIC model to the introduction. This is critical to understanding the paper, and I found myself often confused on how it differs from standard RL models I'm more familiar with.

2) Time correlates of task performance & hippocampal activity.

The authors observe that the global value maximum signal is reflected in AH, while momentary RPEs are reflected in PH. However, it seems like global value maximum signals and RPEs would arise at particular times during the task. Presumably, as subjects learn to predict the global value max, the occurrence of RPEs decreases along with participants' shifts towards exploitation. And of course exploitation can't occur until subjects learn which RT to exploit. So if there was some time-dependent shift in activity across trials that was not specifically related to task performance, would the same results emerge?

Is it really the case, for instance, that AH encoding of global value maximum promotes exploitation, or does learning of the global value max occur in tandem with a behavioral shift from exploration to exploitation as participants learn the best RTs to produce? If AH is reflecting participants' learned values of RT, by the time AH activity has begun to encode the global max, participants are presumably making fewer exploratory choices, which could explain why AH is more correlated with exploitative choices. Perhaps the "unlearnable" variants help sort this out? In any case, some more clarity on these points would be appreciated.

3) The nature of exploratory choices on this task.

It seems like the kind of exploration that PH RPEs promoted was not very useful. If I understand correctly, exploration was not information-seeking about uncertain portions of the RT map, nor truly random, but rather promoted re-checking places already established as low value. It seems like this sort of exploration wouldn't be very useful, and doesn't precisely square with traditional framings of the explore-exploit problem. This also seems different than the rodent optogenetic studies the authors cite, where stimulation promoted exploration of a novel environment, not re-exploring relatively-familiar—though not rewarding—portions of the world. Can the authors comment on the potential adaptive significance of the exploration they observe in the context of the clock task, and how this relates to exploration on other RL tasks?

4) Information compression.

How is the information compression the authors refer to different than learning value of states? I (think) I understand what they authors are describing, but I've never heard this referred to as information compression, though I appreciate it is in a sense. But is information compression as used here different than any sort of error-correcting learning mechanism that assigns a unidimensional value to a particular action or state? If I'm misunderstanding what's happening here, perhaps some additional discussion of the SCEPTIC model and how it contrasts with more familiar RL formulations would be beneficial to general readers.

5) Time vs space.

Time and position in RT space are correlated on this task, since the moving dot sweeps through the state space at a constant speed. Both temporal and spatial information modulate neurons in the rodent hippocampus, and it would be interesting to know which factor value representations are being bound to here. Without new experiments I'm not sure it's possible to extract a definitive answer on this, but it would be interesting to hear the authors' thoughts on which is more meaningful here.

Response to Reviewers for NCOMMS-20-03269: Differential reinforcement encoding along the hippocampal long axis helps resolve the explore/exploit dilemma

Reviewer #1 (Remarks to the Author):

This paper is an important contribution to both the hippocampal literature and the literature on explore-exploit decision making. ... In addition, the manuscript is clearly written, the study well powered (including a behavioral replication), and the data and code are available. We believe it is appropriate for publication in Nature Communications.

We appreciate the reviewers' encouraging comments and helpful suggestions in improving the manuscript.

1. Comparison between findings in this task and previous studies. Michael Frank has published several papers on this task including at least one fMRI paper. How do the whole brain results compare with previously published findings? Of course, one problem here is the different model used by Frank et al. which brings us to a second point.

Like the reviewers, we were very interested in the neural correlates of uncertainty-directed exploration reported by David Badre, Michael Frank and colleagues (2011). However, as we detail in our earlier paper (Hallquist & Dombrovski, 2019, *Cognition*), our effort to reproduce these results encountered several major difficulties. First, the parameters of the 'TC' model were not identifiable ($R^2 < .25$; Supplementary Results; Supplementary Figure 1). Second, in simulated environments the TC model performed barely above chance and was inferior by a large margin to all comparators, including a more robust version of temporal difference (TD) learning (Fig. 4). Third, when the TC model was fit to human behavior, the uncertainty sensitivity parameter (or any substantively interesting parameter of the TC model, Fig. 9) did not contribute to the fit. Fourth, humans are generally uncertainty-averse on the clock task and showed that RT swings, originally described by Michael Frank, are best explained by global uncertainty (e.g. measured by entropy of the SCEPTIC model; Results 4.3.6-4.3.7). This is most consistent with stochastic, not uncertainty-directed, exploration.

Why do our findings diverge? There are several possible reasons. In our study, we collected data from a larger sample ($n=70$ vs. 15) and have used a more representationally powerful and well-identified RL model (detailed in our next response) that was vetted more extensively than the original TC model (e.g., performance in simulated environments). Furthermore, we have performed more nuanced multi-level level analyses of trial-by-trial behavior, including multi-level survival analyses that deal with censoring, an important consideration given that a response ends sampling of the action space. Importantly, we do not take our findings to mean that humans do not engage in uncertainty-directed exploration (e.g., Wilson and colleagues, 2014, provide strong evidence of uncertainty seeking in simpler environments with a longer contingency horizon). It is, however, strongly evident from our analyses that most instances of human exploration on the clock task are stochastic and that if uncertainty is tracked explicitly by subjects, it primarily promotes aversion to uncertain options.

The focus of this paper is on the hippocampal long axis and we are hesitant to provide more detailed descriptions of whole-brain analyses (and their comparisons to Badre et al., 2011) given space limitations and the importance of maintaining a focused narrative. In the interest of completeness, however, we report the results of our whole-brain analyses. To the reviewer's

point, there is overlap between regions identified in our whole-brain entropy map and Badre and colleagues' map for relative uncertainty in the rostralateral prefrontal cortex (Table s2, cluster #7). For reasons detailed above, we cannot be sure that relative uncertainty from the TC model in the analyses of Badre and colleagues is a distinct, behaviorally relevant computational signal separable from global uncertainty. Thus, we interpret the overlap between the maps as evidence that uncertainty-related cognitive demands recruit the rIPFC, in addition to the dorsal attention network. Please also see our response to R2 comment #3.

2. Could the authors briefly discuss differences with and presumably advantages of their model compared to the original model? This may be most appropriate in the methods section, but to readers familiar with the explore-exploit literature, the SCEPTIC model is still new and the benefits for studying exploration in this task would be worth pointing out.

Thank you for this useful suggestion. In the amended manuscript, we have now added a new section entitled, "*Conceptual comparison of SCEPTIC model to earlier TC model.*" In this section, we provide a synopsis of the Frank TC model and also compare it to SCEPTIC, focusing especially on the benefits of SCEPTIC for studying exploration. We provide a relevant excerpt of this section below.

"Previous papers describing behavior on the clock task have suggested that some humans tend to shift toward more uncertain response times⁸⁶ and that this tendency is associated with greater activity in the rostralateral prefrontal cortex². These findings are largely founded on a different computational model of the task, called the TC ('time clock') model, which represents response times on each trial i as a linear combination of several potentially neurobiological processes:

$$\widehat{RT}(i) = K + \lambda RT(i-1) + \nu [RT_{\text{best}} - RT_{\text{avg}}] - \text{Go}(i) + \text{NoGo}(i) + \rho [\mu_{\text{slow}}(i) - \mu_{\text{fast}}(i)] + \epsilon [\sigma_{\text{slow}}(i) - \sigma_{\text{fast}}(i)] \quad (15)$$

The details of each parameter and the underlying representation are provided in previous reports⁸⁶. Briefly, however, with respect to value-based decisions, the TC model separately updates the probability of a positive prediction error for RTs that are slower or faster than the subject's average (μ_{slow} and μ_{fast} , respectively). With learning, the model predicts that subjects shift toward faster or slower RTs that are associated with a greater expectation of a positive prediction errors according to a free parameter, ρ . The definitions of 'fast' and 'slow' responses are based on a comparison to the running average of recent response times. TC tracks the expected value (μ) and uncertainty (σ) using two beta distributions, one for 'fast' and one for 'slow' responses. Our previous computational and behavioral analyses have found that the TC model has problems with parameter identifiability, that its substantive parameters for value and uncertainty do not contribute to model fit in empirical data, and that the model performs poorly in more complex time-dependent contingencies³⁵.

Perhaps more important than these limitations are the conceptual differences between the TC and SCEPTIC models, which render SCEPTIC particularly well-suited for detailed analyses of exploration and exploitation on the clock task. The representation of value over time involves a tradeoff between the generality of representation on one hand and the number of free parameters or values stored on the other. A completely general temporal value representation is exemplified by temporal difference (TD) models, which we have previously tested. On the other end,

parsimonious parametric models such as Frank's TC often turn out to explain a narrow range of phenomena; they break down more easily at boundary conditions.

Radial basis function representation, in our opinion, finds the middle ground between these two extremes: it reduces the memory and computational load compared to TD, while maintaining generality of representation, which enables it to learn virtually any contingency in one continuous dimension. Furthermore, by approximating the value function over the time interval of the task, the SCEPTIC model enables one to test hypotheses about both the chosen action (e.g., its expected value, or reward prediction error) and global statistics such as the entropy of the value function. Moreover, the function approximation approach of SCEPTIC can be extended to test whether humans prefer or are averse to more uncertain options (the Fixed U+V model above). Thus, variants of the SCEPTIC model can disentangle stochastic versus uncertainty-related exploration on the clock task. The former is related to the entropy of learned values that enter into the softmax choice rule; the latter depends on explicit tracking of the sampling uncertainty in a Kalman filter. By comparison, the fast vs. slow parametric representation of TC provides a coarser view of the task that does not distinguish between stochastic and uncertainty-directed exploration and or provide the statistics of the global value maximum."

3. There are some papers that should probably be cited to connect this with the broader explore-exploit literature. Most notably the work by Wu and Schulz who have a few papers on spatially structured explore-exploit tasks with spatial correlations between bandits determined by a Gaussian process. This I believe is the first

Wu, C.M., Schulz, E., Speekenbrink, M. et al. Generalization guides human exploration in vast decision spaces. *Nat Hum Behav* 2, 915–924 (2018).

We thank the reviewer(s) for bringing the work on the Gaussian process models to our attention: these models are indeed mathematically and psychologically homologous to SCEPTIC in that all use radial basis functions (RBF) to reduce the dimensionality of representation. We now acknowledge this work in the revised Introduction and agree on the broader point that humans take advantage of their knowledge of the spatial structure of reward distribution. It is interesting to consider whether the AH could subserve adaptive spatial generalization described by Wu and colleagues (2018). While both models use RBFs to reduce the dimensionality of options, Gaussian process regression focuses on the adaptive kernel whereas SCEPTIC compresses information across RBFs and learning episodes through selective maintenance, a feature specifically decoded from AH responses. It is, however, possible that both formalisms describe the same underlying neural computation.

4. In addition, modern views of explore-exploit decision making focus on two types of exploration - directed (information seeking) and random (behavioral variability). This view was synthesized by

Wilson, R. C., Geana, A., White, J. M., Ludvig, E. A., & Cohen, J. D. (2014). Humans use directed and random exploration to solve the explore-exploit dilemma. *Journal of Experimental Psychology: General*, 143(6), 2074.

But nicely summarized in this review paper by Schulz and Gershman

Schulz, E., & Gershman, S. J. (2019). The algorithmic architecture of exploration in the human brain. *Current opinion in neurobiology*, 55, 7-14.

In this regard, it would be nice to have a paragraph in the discussion connecting the findings to directed and/or random exploration.

As Schulz and Gershman (2019) note, humans switch between uncertainty-driven directed and random exploration strategies depending on the structure of the environment, the distribution of rewards, and the incentives. To this point, we find when the distribution of returns across the environment is reasonably smooth, the more computationally demanding uncertainty-driven exploration does not yield better returns than stochastic exploration (please see our response to R3 comment #3 for details). Further, we have found (Hallquist & Dombrovski, 2019) that exploration on the clock task is predominantly random and that people are generally very uncertainty-averse. This conclusion stands in contradiction with Michael Frank's early findings with this task, and we have documented the reasons for the discrepancy (please see our response to R1 comment #1, and our earlier paper). In this current study, we have examined how hippocampal responses moderated the effects of local uncertainty on choice (Results, AH promotes uncertainty aversion while PH does not modify uncertainty preferences). Following the reviewer(s)' recommendation, we now connect our findings to the directed versus random exploration distinction in the Conclusions (paragraphs 2 and 3).

5. The sample for this study includes a wide age range (from 14 to 30). Previous work on exploration has shown that there are differences between adolescence and young adults in terms of their exploratory behavior, particularly with regards to directed (or exploration driven by information seeking)

Schulz, E., Wu, C. M., Ruggeri, A., & Meder, B. (2019). Searching for rewards like a child means less generalization and more directed exploration. *Psychological science*, 30(11), 1561-1572.

Somerville, L. H., Sasse, S. F., Garrad, M. C., Drysdale, A. T., Abi Akar, N., Insel, C., & Wilson, R. C. (2017). Charting the expansion of strategic exploratory behavior during adolescence. *Journal of experimental psychology: general*, 146(2), 155.

And also differences in ambiguity aversion between teens and young adults

Tymula, A., Belmaker, L. A. R., Roy, A. K., Ruderman, L., Manson, K., Glimcher, P. W., & Levy, I. (2012). Adolescents' risk-taking behavior is driven by tolerance to ambiguity. *Proceedings of the National Academy of Sciences*, 109(42), 17135-17140.

Do you find age differences in terms of behavior within the task? And do patterns of task-related activation differ by age? Regardless of the result, a short paragraph connecting to this previous literature would be good.

We appreciate the reviewer(s) sharing these references. We agree that the results of Schulz et al. (2019) are especially relevant. The current data were collected as part of a larger study of development and the current paper is the initial, normative account of hippocampal substrates of exploration/exploitation. All of our group voxelwise GLM analyses included age and sex as mean-centered covariates such that the maps presented in the Supplement and the regression

coefficients extracted for detailed hippocampal analyses were adjusted for these variables. We plan to report on their developmental trajectories separately and now note this future direction in the revised Conclusions, along with citations to relevant literature. We considered a longer treatment of age-related considerations, but given space constraints and the goal of maintaining the focus of this paper, we elected to defer a deeper consideration of this important question.

“Finally, while our sample varied substantially in age (14-30), we did not test for age-related changes in the hippocampus to exploration or exploitation. This is an important topic for future research given emerging evidence of changes in both exploration^{67,68} and learning from ambiguous and aversive outcomes^{69,70} between adolescence and adulthood.”

In addition, in the methods, we have added a brief summary of age-related differences in task performance and model parameters: “We note that fitted parameters for the SCEPTIC model did not vary significantly as a function of age ($ps > .2$), though overall performance (number of points earned) increased somewhat with age, $r = .29$, $p = .01$.”

Reviewer #2 (Remarks to the Author):

I was very happy to receive the manuscript by Dombrovski and colleagues. The study adds to a growing literature of hippocampal contributions to reinforcement learning, here, specifically exploring the neural mechanisms that underly the arbitration between exploration/exploitation in task that require cognitive mapping of task space. Although there is an extensive literature on how the cerebral cortex and basal ganglia may contribute to this computation, little attention has been directed at the hippocampus, which, based on its computational properties, is well suited to play a role in this tradeoff.

The theoretical justifications for the study are sound, the paradigm is well established, the computational and imaging methods appear rigorous and appropriate, the study is well powered (including out of session replication). The results, I believe, will also be of great interest to the wider field. In addition I find the writing clear.

Thank you for these positive remarks about our study.

1. Can the authors please define 'RT swing' more clearly. For being such an important regressor I was surprised that I could not find exactly how it was computed. Is it simply the absolute RT difference between RT on trial N and trial N-1?

As we now clarify in the revised Results (Posterior hippocampal responses to local reinforcement promote exploration) and Methods (Analyses of behavior using frequentist multilevel models/Multilevel regression models), our analyses define RT swings as the trial-level effect of a given RT on the next RT (autocorrelation) in the trial-wise multilevel models. A weaker effect indicates greater RT swings.

While the absolute RT difference between consecutive trials is an intuitive metric, it suffers from several measurement problems. First, it has an inherently zero-inflated distribution and cannot be treated as approximately normally distributed. Second, due to time-varying imprecision, this absolute difference scales with the RTs. Third, it depends on where the preceding RT is relative to the edge of the interval.

2. In the section 'Voxelwise general linear model analyses' the authors describe that only one model-based regressor was added at a time in the parametric fMRI models. It is reassuring that the results are allegedly quantitatively similar when all SCEPTIC regressors were included, however, since these analyses provide the basis for the main results it would be good to see this elaborated on in the paper with additional analysis (perhaps supplemental) rather than simply upon request. I do not know what "relatively low correlation" means in this context and I would like to be reassured that the main results survives taking all the correlated regressors into account.

Thank you for drawing our attention to this consideration. We agree that it is important for the main results to survive when correlated regressors are included in the model. Our primary analyses in the paper examine the effects of reward prediction errors and the trial-wise entropy of the learned value distribution (a measure of the prominence of the global value maximum). Thus, we now provide additional details on the correlation between these regressors in fMRI in the supplement. Furthermore, we display the key double dissociation between (low) entropy and reward prediction errors (Fig. 2b) based on individual versus simultaneous fMRI GLM models in Supplementary Figure S4. The amended methods section now reads:

“We further verified that the key double dissociation between prediction errors and entropy along the long axis of the hippocampus (Fig. 2) held when entropy and reward prediction errors were included simultaneously in the fMRI GLMs. As shown in Fig. S4, there was no meaningful difference in the double dissociation when GLM coefficients were extracted from models with one model-based regressor each (i.e., separate models for entropy and prediction errors) versus a model that included both of these regressors simultaneously.

“Importantly, the results of these and other fMRI analyses would only diverge if the model-based regressors had a moderate to strong correlation with each other, leading to collinearity problems. To examine this possibility, we computed the correlation between the convolved regressors for reward prediction errors and entropy for all subjects and runs. We then modeled the correlation in a Bayesian multilevel model (implemented in the *brms R* package⁸²) that included a random intercept of subject and allowed for heterogeneity between runs in the variability of the RPE-entropy correlation. This analysis revealed a very small average correlation between RPE and entropy, $r = 0.07$, 95% highest posterior density interval = .05 – .09. Following Cohen's rules of thumb, we further tested for the probability that the PE-entropy correlation is small, $|r| < .10$ using a region of parameter equivalence (ROPE) test on the posteriors from the Bayesian multilevel model. This test revealed that 100% of the posterior samples of the PE-entropy correlation fell within this range, providing strong evidence that the correlation between entropy and PE is small. Altogether, the low level of correlation between these convolved model-based signals indicates that any additional analyses based on regression coefficients from the fMRI GLMs would be very similar regardless of whether the signals were modeled individually or simultaneously, consistent with Fig. S4.”

3. As far as I understand the learning rate is a free parameter in the SCEPTIC model, however, the trial-level alternative model of reinforcement learning, learning rate alpha is set to .1. Does the supremacy of SCEPTIC hold across the whole range of learning rates in the simpler model?

As you noted, the learning rate in the SCEPTIC model is a free parameter, which could give it an unfair advantage relative to a trial-level fixed learning rate model. Following your suggestion, we fit the trial-fixed model to behavior using learning rates of $\alpha = .05, .10, .15, .20$, then refit the voxelwise GLMs using the corresponding prediction error regressors. We did not observe any qualitative change in the superiority of the SCEPTIC PE regressor relative to the fixed model in representing posterior hippocampal activity. Rather, in middle slices of the hippocampus, the superiority of SCEPTIC PEs relative to fixed-trial learning models increased for higher learning rates. For example, PE modulation in hippocampal slices 4 and 5 was significantly stronger for SCEPTIC PEs relative to the trial-level $\alpha = .20$ model.

The detailed results of these multilevel analyses are presented below. Each row represents a test of a SCEPTIC PE – trial-fixed PE difference in axis bins from posterior (1) to anterior (12). The p-values are adjusted for all slices simultaneously using a single-step multivariate method (Hothorn and colleagues). In the amended manuscript, we now summarize that the superiority of SCEPTIC PE representation held over learning rates between .05 and .2 for the trial-level model.

$\alpha = .05$

Linear Hypotheses:

	Estimate	Std. Error	z value	Pr(> z)	
1 == 0	0.0655126	0.0178749	3.665	0.002963	**
2 == 0	0.0740268	0.0182827	4.049	0.000617	***
3 == 0	0.0519173	0.0176177	2.947	0.037846	*
4 == 0	0.0438045	0.0191898	2.283	0.238490	
5 == 0	0.0378492	0.0176177	2.148	0.320481	
6 == 0	-0.0024596	0.0177449	-0.139	1.000000	
7 == 0	-0.0125584	0.0181437	-0.692	0.999682	
8 == 0	-0.0118531	0.0180078	-0.658	0.999810	
9 == 0	-0.0018785	0.0182827	-0.103	1.000000	
10 == 0	0.0004984	0.0190292	0.026	1.000000	
11 == 0	0.0198587	0.0176177	1.127	0.972886	
12 == 0	0.0228147	0.0182827	1.248	0.942743	

(Adjusted p values reported -- single-step method)

$\alpha = .10$

Linear Hypotheses:

	Estimate	Std. Error	z value	Pr(> z)	
1 == 0	0.066582	0.017854	3.729	0.0023	**
2 == 0	0.071837	0.018261	3.934	0.0010	**
3 == 0	0.053034	0.017597	3.014	0.0305	*
4 == 0	0.046443	0.019167	2.423	0.1699	
5 == 0	0.041854	0.017597	2.378	0.1898	
6 == 0	0.007047	0.017724	0.398	1.0000	
7 == 0	-0.005015	0.018122	-0.277	1.0000	
8 == 0	-0.009611	0.017987	-0.534	1.0000	
9 == 0	-0.001064	0.018261	-0.058	1.0000	

10 == 0	-0.001819	0.019007	-0.096	1.0000
11 == 0	0.014381	0.017597	0.817	0.9984
12 == 0	0.014326	0.018261	0.784	0.9989

(Adjusted p values reported -- single-step method)

$\alpha = .15$

Linear Hypotheses:

	Estimate	Std. Error	z value	Pr(> z)	
1 == 0	0.069513	0.017844	3.896	0.001175	**
2 == 0	0.071971	0.018251	3.943	0.000964	***
3 == 0	0.057088	0.017587	3.246	0.013958	*
4 == 0	0.052249	0.019157	2.727	0.073969	.
5 == 0	0.049125	0.017587	2.793	0.060861	.
6 == 0	0.017956	0.017714	1.014	0.988505	
7 == 0	0.004932	0.018113	0.272	1.000000	
8 == 0	-0.002476	0.017977	-0.138	1.000000	
9 == 0	0.002718	0.018251	0.149	1.000000	
10 == 0	-0.002246	0.018997	-0.118	1.000000	
11 == 0	0.010385	0.017587	0.590	0.999940	
12 == 0	0.009358	0.018251	0.513	0.999987	

(Adjusted p values reported -- single-step method)

$\alpha = .20$

Linear Hypotheses:

	Estimate	Std. Error	z value	Pr(> z)	
1 == 0	0.0728349	0.0178451	4.082	0.000537	***
2 == 0	0.0729769	0.0182523	3.998	0.000766	***
3 == 0	0.0617362	0.0175883	3.510	0.005363	**
4 == 0	0.0591949	0.0191579	3.090	0.023769	*
5 == 0	0.0575697	0.0175883	3.273	0.012687	*
6 == 0	0.0289556	0.0177153	1.634	0.725579	
7 == 0	0.0156996	0.0181135	0.867	0.997134	
8 == 0	0.0076443	0.0179778	0.425	0.999998	
9 == 0	0.0099335	0.0182523	0.544	0.999975	
10 == 0	0.0006562	0.0189976	0.035	1.000000	
11 == 0	0.0103975	0.0175883	0.591	0.999939	
12 == 0	0.0082794	0.0182523	0.454	0.999997	

(Adjusted p values reported -- single-step method)

4. I found it very interesting that PH RPE responses loaded on a factor distinct from all other whole-brain-significant RPE-sensitive regions, while low-entropy AH responses were on the same factor as ventromedial prefrontal cortex. What does this mean for the double dissociation and how should this be interpreted in terms of neural mechanisms supported by the hippocampus and other brain regions? possible to say anything about mechanism given this result? Could it imply that PH RPE responses are simply inherited from other

brain regions while there is some representation of value of states in AH that are computed there? I wouldn't mind some additional elaboration on the implications of this result.

This is an interesting suggestion, and the broader question posed by the reviewer is important. The between-subject correlations among regional responses ("betas") are a coarse measure of functional co-activation, which we report for descriptive purposes, given that these signals are novel. We reported the most parsimonious and interpretable factor solutions for each analysis but are not confident enough to make inferences based on the number and structure of factors. The functional separability of AH and PH from other co-active regions would require more focused connectivity analyses. One explanation for this loading structure is that all RPE-responsive regions displayed positive responses, allowing more than one RPE-positive factor to emerge, while entropy elicited both negative and positive regional responses, with all positive regions loading on the same factor. To the reviewer's next point, we consider within-subject connectivity indices, such as those estimated by dynamic causal modeling or psychophysiological interaction analyses, to support stronger inference about functionally inter-connected networks.

On a related note, did the authors you ever attempt causal modelling to try and disambiguate whether the described regressors are computed in the hippocampus or elsewhere (I'm not saying that causal modelling is necessary but asking out of interest)?

This is an excellent suggestion. We are in fact currently working on dynamic causal modeling analyses examining the direction of anterior hippocampus – vmPFC effective connectivity related to the signaling of the global value maximum statistics. We are also planning a similar analysis of striatal-posterior hippocampus connectivity related to RPEs. These analyses may provide more conclusive evidence about how hippocampus integrates with other brain regions to support exploration and exploitation.

—

Reviewer #3 (Remarks to the Author):

In this manuscript, the authors examine hippocampal responses on a reinforcement learning task, with a particular interest in understanding the explore-exploit trade off. Using a clever behavioral task, they had participants learn different response strategies suitable for different distributions of expected value across strategy space. They report a double dissociation of hippocampal activity, with posterior hippocampus encoding moment by moment reward prediction errors as subjects learn the best response strategy, and anterior hippocampus encoding a representation of value over a longer time horizon.

I liked the way the authors frame this task as a continuous, spatially-structured complement to bandit tasks with discrete options. The goal of extending the reinforcement learning framework from experimentally-tractable but less realistic situations to problems mimicking the complex spatial and temporal dependencies in the real work make this work timely and interesting, in my opinion. Overall, I found the paper well written, if a bit difficult to parse in some areas. However, I think the analyses and results are compelling, and are a valuable contribution to the literature implicating hippocampus in RL processes.

We appreciate the encouraging remarks about the paper and have worked to make it easier to parse, following your detailed recommendations below.

1. As a general comment, I'd suggest adding more explanatory information about the task and especially the SCEPTIC model to the introduction. This is critical to understanding the paper, and I found myself often confused on how it different from standard RL models I'm more familiar with.

We appreciate the importance of providing additional details about the task and model. As you note, the model is relatively new and its links with standard RL models need to be explicated. In the Methods, we now provide a detailed comparison of the SCEPTIC model with the TC model (Frank and colleagues), highlighting the value of our model for studying the explore/exploit dilemma (cf. our response to R1, comment #2 above).

In the amended introduction, we now note the relationship between SCEPTIC and standard RL models. The main technical innovation of our model is the application of temporal basis functions to learn approximate the value function over the four-second interval of the clock task. Furthermore, we have previously found (Hallquist & Dombrovski, 2019, *Cognition*) that a SCEPTIC variant that selectively maintains chosen actions while allowing the traces of unchosen values to decay yields an information-compressing model that provides a superior fit to human behavior relative to alternatives (including variants that explicitly track uncertainty). We now note these points in the introduction. Finally, we now provide additional model details in the opening of the Results (SCEPTIC reinforcement learning model captures local and global reinforcement).

2) Time correlates of task performance & hippocampal activity.

The authors observe that the global value maximum signal is reflected in AH, while momentary RPEs [reward prediction errors] are reflected in PH. However, it seems like global value maximum signals and RPEs would arise at particular times during the task. Presumably, as subjects learn to predict the global value max, the occurrence of RPEs decreases along with participants' shifts towards exploitation. And of course exploitation can't occur until subjects learn which RT to exploit. So if there was some time-dependent shift in activity across trials that was not specifically related to task performance, would the same results emerge?

Is it really the case, for instance, that AH encoding of global value maximum promotes exploitation, or does learning of the global value max occur in tandem with a behavioral shift from exploration to exploitation as participants learn the best RTs to produce? If AH is reflecting participants' learned values of RT, by the time AH activity has begun to encode the global max, participants are presumably making fewer exploratory choices, which could explain why AH is more correlated with exploitative choices. Perhaps the "unlearnable" variants help sort this out? In any case, some more clarity on these points would be appreciated.

In our view, this was an important unanswered empirical question about our findings. We thank the reviewer for prompting us to examine more thoroughly the time course of model-predicted signals mapped to the hippocampus. Before turning to these new results, however, three trivial explanations for the dissociation of responses in AH vs. PH can be ruled out. First, a simple effect of time (e.g., linear drift) is unlikely to explain these effects because BOLD data were detrended using a .008 Hz temporal high-pass filter. Second, RPEs do not asymptote late in learning on the

clock task (see plots below) because all contingencies are probabilistic and the point magnitudes or rewards vary substantially. Third, as we note in our response to R2's comment #2, RPEs and entropy are uncorrelated and thus did not simply index alternate epochs in the experiment.

To the reviewer's point, we agree with R1's assessment that the difference between posterior and anterior hippocampal responses is in the timescale of modulation, a point we now make in the revised Conclusions (para 1). PH responds to trial-by-trial changes in reinforcement and AH responses evolve more slowly (but not linearly) over the course of learning (run; see Fig. 7 in Hallquist & Dombrovski, 2019, included below). The critical question, however, is whether AH responses specifically reflect the evolution of value entropy vs. a more general shift from the unfamiliar environment encountered early in the run to the extensively sampled environment late in the run.

M.N. Hallquist, A.Y. Dombrovski

Cognition 183 (2019) 226–243

Fig. 7. Evolution of value entropy starting from random uniform prior estimates on value. Lines represent the mean entropy across trials, averaging across subjects. Trial-wise entropy was derived from the estimated value distributions of subjects at their best-fitting parameters. Shaded ribbons represent the bootstrapped 95% confidence interval of the mean at each trial. In panel a, dark vertical lines depict boundaries between different contingencies (50 trials each), explicitly signaled to participants. Panel b depicts the average change in entropy, averaging over subjects and runs (excluding run 1); this represents the typical increase in entropy during initial exploration followed by its decline as high-value actions are discovered and exploited. Better-performing subjects (right panel) exhibit proportionately greater entropy increases early in learning under the selective maintenance model, whereas poorer subjects (left panel) have higher mean entropy. Value traces were carried forward from one block to the next, an implementation that resulted in better fits for both models compared to resetting values in each block (data available upon request). Apart from differences in the first few trials of the experiment, the essential dynamics of entropy under the Fixed LR V and Selective Maintenance models are unchanged if the model is initialized with zero prior estimates on value (see Supplementary Fig. 6).

To answer this question definitively, we have now split every run into its first and second halves. If indeed PH were active early in learning under non-specific demands other than RPEs and AH were active late in learning under demands other than decreasing value entropy, both signals would be abolished or at least considerably attenuated in split data, which eliminate the contrast between early and late epochs. Alternatively, if PH responded selectively to trial-wise RPEs and AH responded selectively to fluctuations of entropy, both signals would persist in split data. In fact, both signals persisted in these admittedly under-powered analyses. If anything, the long axis gradient for RPE signals was more pronounced late in learning. Given that entropy signals evolve more slowly, we were less certain about what to expect. We were very reassured by the

persistence of both the AH entropy signals and the AH/PH dissociation in both halves. We note that the low entropy responses in the AH body are somewhat stronger in the first vs. second half of each run, which is not surprising given the greater variability in entropy observed in trials 1-20 (reprinted Fig. 7 above). The split-half fMRI analysis is now reported in the Results:

“Critically, PH RPE and AH global value responses were not an artifact of novelty or some other time-dependent shift in activity unrelated to exploration/exploitation, as these signals persisted when early and late parts of each run were analyzed as separate regressors. More specifically, when we extracted GLM regression coefficients in the hippocampus from regressors representing the first and second halves of the task, the double dissociation between PH RPE and AH global value responses held in both the first half (trials 1-25; anteroposterior location \times signal $\chi^2(11) = 1361.03$, $p < 10^{-16}$) and second half (trials 26-50; anteroposterior location \times signal $\chi^2(11) = 1685.54$, $p < 10^{-16}$). In a model that treated run half as a categorical moderator, we found an anteroposterior location \times signal \times half interaction, $\chi^2(11) = 192.50$, $p < 10^{-16}$, such that entropy modulation was more pronounced in mid-anterior slices early than late in learning, while RPEs became more focally associated with positive PH modulation late in learning (see Fig. S2)”.

Figure S2.

3) The nature of exploratory choices on this task.

It seems like the kind of exploration that PH RPEs promoted was not very useful. If I understand correctly, exploration was not information-seeking about uncertain portions of the RT map, nor truly random, but rather promoted re-checking places already established as low value. It seems like this sort of exploration wouldn't be very useful, and doesn't precisely square with traditional framings of the explore-exploit problem. This also seems different than the rodent optogenetic studies the authors cite, where stimulation promoted

exploration of a novel environment, not re-exploring relatively-familiar—though not rewarding—portions of the world. Can the authors comment on the potential adaptive significance of the exploration they observe in the context of the clock task, and how this relates to exploration on other RL tasks?

Our characterization of PH RPE-enhanced RT swings on the clock task was insufficiently clear. As we note in our response to R1's comment about (uncertainty-)directed vs. random exploration, RT swings on the clock task represent mostly random exploration, approximated by the softmax choice rule. Our survival analyses showed that PH RPEs promoted visits to regions with *relatively* low value. Interpreted in the context of a main effect of value on choice, however, this behavior is better described as more random sampling of alternative RTs with a bias toward valuable RTs (i.e., hotter softmax choice) rather than a true preference for lower values.

To the reviewer's substantive point, the literature on human exploration indeed emphasizes uncertainty-directed strategies and sometimes suggests them to be superior. Computational studies reported in our earlier paper suggest that this not necessarily the case in the complex environment of the clock task. Across a range of contingencies, softmax exploration was not inferior to more cognitively demanding uncertainty-directed exploration: while the former recovered the contingency more accurately, it did not improve returns over horizons of 40-110 trials. Uncertainty-directed exploration became only modestly advantageous solely in "Easter egg" environments with very sparse, non-smooth distributions of returns (Hallquist & Dombrovski, *Cognition*, 2019, Fig. 4 and s3-s4).

We now comment on the nature of PH-driven exploration in the revised Conclusions (para 2).

4) Information compression.

How is the information compression the authors refer to different than learning value of states? I (think) I understand what they authors are describing, but I've never heard this referred to as information compression, though I appreciate it is in a sense. But is information compression as used here different than any sort of error-correcting learning mechanism that assigns a unidimensional value to a particular action or state? If I'm misunderstanding what's happening here, perhaps some additional discussion of the SCEPTIC model and how it contrasts with more familiar RL formulations would be beneficial to general readers.

As the reviewer notes, error-correcting learning mechanisms compress information across episodes (here, trials). In the revised Introduction and Methods we clarify that, in addition, the SCEPTIC model compresses information across the available actions within a trial, here represented by radial basis function elements that cover the continuous action space. This is accomplished by selectively maintaining values of elements that represent preferred actions and allowing others to decay (Equation 6). As we describe in the Results and detail in our earlier paper, these dynamics accentuate the learned value maximum, erase differences between the values of other, inferior actions, and accelerate the transition from exploration to exploitation. As we report in our earlier paper (Hallquist & Dombrovski, 2019; Fig. 7b, reproduced above), late in learning, selective maintenance reduced memory requirements by 1/3 in poorly performing subjects and by almost 1/2 in well-performing subjects.

5) Time vs space.

Time and position in RT space are correlated on this task, since the moving dot sweeps through the state space at a constant speed. Both temporal and spatial information modulate neurons in the rodent hippocampus, and it would be interesting to know which factor value representations are being bound to here. Without new experiments I'm not sure it's possible to extract a definitive answer on this, but it would be interesting to hear the authors' thoughts on which is more meaningful here.

We agree with the reviewer that, since both place cell representations and the long-axis gradient in the hippocampus are domain-general, one would expect the current results to generalize to spaces with dimensions other than time. What makes time special in a human experiment is that its unidirectional flow and the resulting time pressure make it hard for subjects to respond strategically and force them to rely on trial-by-trial associative learning. Our revised Conclusions discuss future experiments that would establish domain-generality of the long-axis gradient in the context of exploration/exploitation. We are testing this hypothesis in a follow-up study with continuous and discrete non-temporal environments.

REVIEWERS' COMMENTS:

Reviewer #1 (Remarks to the Author):

The authors have addressed my concerns. This will be a great addition to the literature.

Reviewer #2 (Remarks to the Author):

The authors have adequately addressed my concerns and I recommend that the manuscript is accepted for publication.

Reviewer #3 (Remarks to the Author):

No further questions!